# FOUNDER: Grounding Foundation Models in World Models for Open-Ended Embodied Decision Making

Yucen Wang [1 2]   Rui Yu [1 2]   Shenghua Wan [1 2]   Le Gan [1 2]   De-Chuan Zhan [1 2]

## Abstract

Foundation Models (FMs) and World Models (WMs) offer complementary strengths in task generalization at different levels. In this work, we propose `FOUNDER`, a framework that integrates the generalizable knowledge embedded in FMs with the dynamic modeling capabilities of WMs to enable open-ended task solving in embodied environments in a reward-free manner. We learn a mapping function that grounds FM representations in the WM state space, effectively inferring the agent's physical states in the world simulator from external observations. This mapping enables the learning of a goal-conditioned policy through imagination during behavior learning, with the mapped task serving as the goal state. Our method leverages the predicted temporal distance to the goal state as an informative reward signal. `FOUNDER` demonstrates superior performance on various multi-task offline visual control benchmarks, excelling in capturing the deep-level semantics of tasks specified by text or videos, particularly in scenarios involving complex observations or domain gaps where prior methods struggle. The consistency of our learned reward function with the ground-truth reward is also empirically validated. Our project website is https://sites.google.com/view/founder-rl.

## 1. Introduction

Foundation Models (FMs), pre-trained on diverse datasets, encapsulate rich and generalizable knowledge, demonstrating exceptional capabilities in vision and language tasks (Moroncelli et al., 2024). However, adapting FMs to em-

bodied domains poses significant challenges. Reinforcement Learning (RL) offers a pathway to bridge this gap by enabling FMs to learn and adapt through real-world interactions and feedback. In turn, FMs provide agents with valuable reasoning and generalization abilities. This synergy between RL and FMs has driven extensive research aimed at advancing embodied decision-making (Brohan et al., 2023b; Du et al., 2023; Brohan et al., 2023a; Liu et al., 2024).

Simultaneously, World Models (WMs) have emerged as critical components for embodied RL agents (Wu et al., 2022; Zhang et al., 2024), which capture the underlying dynamics of physical environments and provide actionable state representations essential for control tasks. WMs enable agents to generate synthetic trajectories through imagination, significantly improving sample efficiency (Moerland et al., 2023; Luo et al., 2022). Furthermore, the ability of WMs to model environment dynamics highlights their potential for task generality, as the same learned dynamics model can be applied across multiple tasks within the environment. However, leveraging WMs for specific tasks typically requires the design of tailored reward functions, which can be labor-intensive and challenging. This constraint limits the broader applicability of WMs, especially for tasks specified in human-intuitive formats, such as text descriptions or videos, where rewards cannot be directly inferred.

Since WMs are primarily trained on embodied interaction data, they inherently lack the prior knowledge and high-level semantic understanding required to interpret such open-ended tasks. However, this gap can be filled by FMs, which exhibit powerful capabilities in understanding and representing complex patterns across multiple modalities. This raises a fundamental yet unexplored question: Can the complementary strengths in task generalization offered by WMs (low-level dynamics modeling of the target environment) and FMs (high-level knowledge of the world) be integrated to enable embodied decision-making agents capable of solving diverse, open-ended tasks?

In this work, we propose a framework, *Foundation mOdels groUNDed in woRld modEls for open-ended decision making* (`FOUNDER`), that bridges the gap between FMs and WMs by grounding FM-generated task representations

[1]School of Artificial Intelligence, Nanjing University, China [2]National Key Laboratory for Novel Software Technology, Nanjing University, China. Correspondence to: De-Chuan Zhan <zdc@nju.edu.cn>.

*Proceedings of the 42nd International Conference on Machine Learning*, Vancouver, Canada. PMLR 267, 2025. Copyright 2025 by the author(s).

into actionable WM states, enabling model-based Goal-Conditioned Reinforcement Learning (GCRL) to solve open-ended tasks indicated by texts or videos. While FMs can handle multi-modal task prompts, the output general-purpose representations may not align well with the target RL environment, making them unsuitable for direct use in WMs or policy learning. To address this issue, we ground FM representations within the target environment by learning a mapping function from FM representations to states in the WM. This can be seen as identifying the underlying physical states in the environment simulator that correspond to external observations, represented as FM-generated visual or language embeddings, with the learned WM serving as an environment simulator. The mapping allows multi-modal task prompts to be translated into the WM state space, where they can be treated as goal states. Solving the task then reduces to efficient GCRL in the WM. Specifically, we learn the mapping function using an autoencoder, as well as enforcing alignment between the mapped states and their corresponding WM states using KL constraints. Additionally, we learn a model to predict the temporal distance between WM states, with the predicted distance to the mapped goal state serving as the reward function for policy learning.

We evaluate FOUNDER on DeepMind Control Suite tasks, Franka Kitchen, and Minecraft. Our method consistently performs well in multi-task decision-making based on offline reward-free RL from visual observations with tasks specified in multi-modal prompts. Furthermore, empirical studies in cross-domain settings demonstrate the superior ability of FOUNDER to capture deep-level task semantics, especially in scenarios where previous methods struggle.

The most similar prior method, GenRL (Mazzaglia et al., 2024), relies on step-by-step visual alignment in connecting FM representations with WMs as well as in policy learning. While effective in some cases, this approach may cause misleading task interpretations and reward signals, particularly in tasks with complex visual observations, inherent static nature, or significant visual discrepancies with the prompt. This is because GenRL uses only visual features to infer and match specific agent behaviors, lacking temporal awareness and a deeper understanding of the task semantics.

In contrast, FOUNDER enables a deeper task interpretation by mapping the prompt representation to the corresponding goal state in the WM. This mapping, combined with our temporal-distance-based approach, yields a highly informative reward function that effectively guides policy learning. Unlike GenRL's rigid step-by-step trajectory matching, FOUNDER operates within a more flexible GCRL paradigm, allowing for more adaptive behavior learning. The consistency of our learned reward function with the ground-truth reward is also empirically validated. From those sides, FOUNDER creates a bridge between high-level knowledge

and low-level dynamics, offering a general framework for open-ended task interpretation and accomplishment.

## 2. Related Work

**Task/Reward Specification using Foundation Models**
Foundation models have proven to be powerful tools for task and reward specification in RL. Large language models have been used to design reward functions and apply regularization for improved RL performance (Kwon et al., 2023; Hu & Sadigh, 2023; Klissarov et al., 2023). Vision-language models, in particular, excel in reward and task specification in visually rich environments. Some works (Fan et al., 2022; Sontakke et al., 2024; Baumli et al., 2023) compute semantic similarity between agent states and task descriptions, providing dense reward signals for RL from visual observations. Other studies (Lee et al., 2023) leverage Vision-language models to analyze visual inputs and offer preference-based feedback for reward model training. While some approaches (Baumli et al., 2023; Rocamonde et al., 2023; Cui et al., 2022) use off-the-shelf FMs for zero-shot task specification, others (Fan et al., 2022; Sontakke et al., 2024) fine-tune the FMs on domain-specific datasets for better reward design. Our method differs from these previous works by specifying tasks and rewards through world models.

**Model-Based Goal-Conditioned RL (MBGCRL)** Previous methods have explored GCRL in WMs. LEXA (Mendonca et al., 2021) is an unsupervised RL method that encourages agents to discover and achieve complex goals in imagination, enabling zero-shot adaptation to visual goals. Choreographer (Mazzaglia et al., 2022) exploit the learned WM to learn skills from an offline dataset and uses a meta-controller to deploy them for efficient task adaptation. Director (Hafner et al., 2022) decomposes long-horizon tasks into goals and achieves them in imagination. These methods use MBGCRL for exploration or subtask learning, whereas we apply it directly for behavior learning to reach goal states specified by multi-modal prompts.

## 3. Problem Setting

We focus on training agents capable of solving open-ended tasks in the context of offline RL from visual observations. The agent has access to a pre-collected dataset of interaction trajectories in an embodied environment, denoted as $\mathcal{B} = (o_{0:T}^j, a_{0:T}^j)_{j=1}^N$, where $o$ represents the observations (images) and $a$ represents the corresponding actions. Notably, the dataset $\mathcal{B}$ consists solely of observation-action pairs, and does not contain reward information or task-specific label annotations.

In this setting, the offline RL agent is tasked with addressing open-ended tasks specified by multi-modal prompts, such

as text descriptions or videos. We leverage a Foundation Model, i.e., a Video-Language Model (VLM), to aid the agent in interpreting the task semantics from the prompt. The agent cannot interact with the real environment or obtain the ground-truth reward of the task during training. Thus, the agent's objective is to use the VLM to understand the task, derive the necessary supervisory signals for accomplishing it, and utilize the reward-free offline dataset $\mathcal{B}$ to learn a policy conditioned on the task prompt.

# 4. Method

Figure 1 illustrates the key steps of FOUNDER. In the pre-training stage, we first train a Dreamer-style world model using the offline data. Next, we employ this world model to learn a temporal distance prediction model between world model states. We then learn a mapping function that connects VLM representations to model states, optimizing the reconstruction likelihood of an autoencoder subject to KL divergence constraints. Finally, for downstream tasks specified by text or video prompts, we utilize the components developed in the previous phases to perform behavior learning. We then describe our method in detail.

## 4.1. World Model Learning

We train the world model in the paradigm of DreamerV3 (Hafner et al., 2023) and employ the RSSM backbone (Hafner et al., 2019) from the offline dataset, consisting of the following components:

$$
\begin{aligned}
\text{Posterior (Encoder):} \quad & q_\phi(z_t \mid o_{\leq t}, a_{<t}) \\
\text{Prior (WM transition):} \quad & p_\phi(z_t \mid z_{t-1}, a_{t-1}) \quad (1) \\
\text{Observation Decoder:} \quad & p_\phi(o_t \mid z_t)
\end{aligned}
$$

The learning objective is optimizing the evidence lower-bound on the log-likelihood of the offline data $(o_t, a_t)_{t=1}^T$:

$$
\sum_{t=1}^{T} \mathbb{E}_{z_t \sim q_\phi} \Big[ \ln p_\phi(o_t \mid z_t) - \\
\mathbb{D}_{\text{KL}} \left[ q_\phi(\cdot \mid o_{\leq t}, a_{<t}) \,\|\, p_\phi(\cdot \mid z_{t-1}, a_{t-1}) \right] \Big] \quad (2)
$$

There is no reward model since the dataset contains no reward information. The WM state $z_t$ is composed of both deterministic and stochastic states: $z_t = (h_t, s_t)$. In GenRL, the encoder and decoder do not condition on $h_t$ which contains rich historical information, and the latent states only contain information about a single visual observation (Mazzaglia et al., 2024). This limits GenRL to visual-level multi-step alignment, where FM representations are aligned solely with the visual states $s_t$ in the WM. In contrast, we maintain the original structure of DreamerV3, preserving all the information in the WM states. This ensures that the learned WM retains richer environmental information, allowing for the

learning of a more effective mapping function that captures deeper environmental attributes from the VLM to the WM.

## 4.2. Mapping Function Learning

The learned world model can be viewed as an environment simulator. Intuitively, when a task is specified by a video, the video embedding $e$ generated by a Foundation Model can be considered an external observation of a specific agent state $z$ in the environment: $e \sim \mathcal{U}(\cdot \mid z)$, where $\mathcal{U}$ is the emission function in the context of POMDP (Kaelbling et al., 1998). This is because task-related semantics, which specify a particular agent behavior, are embedded within the video; however, the underlying physical state information is not inherently included in the original representation. Thus, learning the mapping function $q$ from the FM representation space $E$ to the world model state space $Z$, is essentially the process of inferring the corresponding agent state in the simulator: $\hat{z} \sim Q(\cdot \mid e)$.

These insights are reminiscent of prior work on inferring latent states in a pre-defined domain from external visual observations (Chen et al., 2021), which employs variational inference when paired data from the two domains are unavailable. However, given the availability of a Video-Language Model, a pre-trained world model, and an offline dataset, we can construct paired data for the VLM and WM space, using the dataset trajectories. This is done by processing consecutive observation sequences (short videos) in the dataset trajectories using the VLM and encoding the image-action pairs to latent states using the WM, where $k$ is the number of video frames that the VLM can process:

$$
\begin{aligned}
e_t &= \text{VLM}(o_{t-k:t}), \quad t = k, \cdots, T \\
z_t &\sim \text{WM}(\cdot \mid o_{\leq t}, a_{<t}), \quad t = k, \cdots, T
\end{aligned} \quad (3)
$$

Here, $z_t$ is sampled from the encoded distribution of the WM states, capturing the agent's behavior by incorporating historical trajectory information. $e_t$ typically represents the semantics of the recent image sequence up to step $t$ from the perspective of the VLM. The paired data $(e_t, z_t)$ establish the correspondence between the VLM and WM representation spaces. We then learn a mapping function $Q_\psi$ to obtain the mapped state $\hat{z}_t$ by aligning the distribution of the mapped states with the ground-truth states:

$$
\min_{Q_\psi} \sum_{t=1}^{T} \mathbb{D}_{\text{KL}} \left[ Q_\psi(\cdot \mid e_t) \,\|\, \text{WM}(\cdot \mid o_{\leq t}, a_{<t}) \right] \quad (4)
$$

The distribution of $z_t$ is parameterized in the world model, and we similarly parameterize the distribution of the mapped result $\hat{z}_t$. This makes the objective of minimizing the KL divergence tractable, and $Q_\psi$ can be updated using the reparameterization trick. Additionally, since $e_t$ captures the semantics of a short video, including timesteps preceding

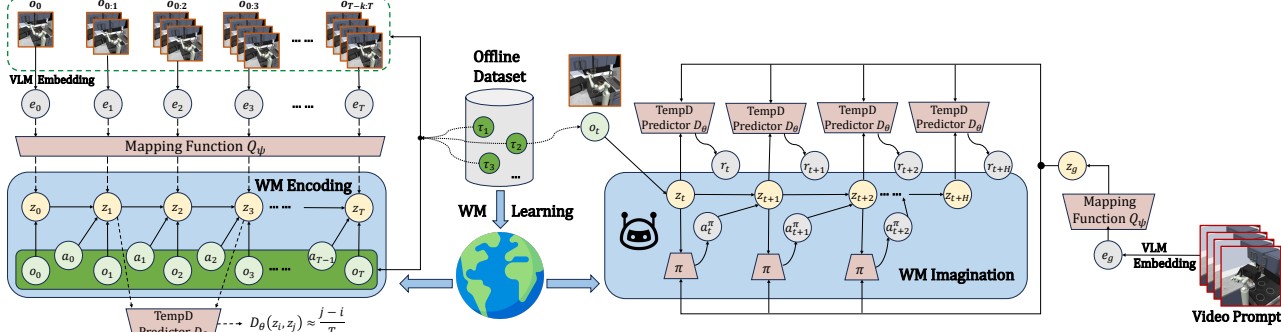

*Figure 1.* FOUNDER encompasses pretraining phase and behavior learning phase. During pretraining, we first train a Dreamer-style WM, then use it to learn a mapping function and a temporal distance predictor. During behavior learning, the VLM representation of a task prompt is mapped to a goal state within the WM and a goal-conditioned policy is learned through imagination, where the predicted temporal distance to the goal state serves as reward. The three pretrained components collectively enable successful behavior learning.

$o_t$, aligning only the last step's states may overlook prior information. To avoid seq2seq-style modeling, which requires learning a complex generative mapping function as in GenRL, we instead introduce a simple reconstruction loss to recover the complete VLM representation from the mapped state, using an autoencoder:

$$\min_{Q_\psi, P_\psi} \quad \mathbb{E}_{\hat{z}_t \sim Q_\psi(\cdot|e_t)} \left[ -\ln P_\psi \left( e_t \mid \hat{z}_t \right) \right] \quad (5)$$

The two losses together result in an objective akin to variational inference, with the key difference being that we use the alignment of WM states at the corresponding timesteps as a KL constraint, rather than relying on random state distributions. Although the mapping is learned solely on interaction data from the target environment, the generalization power of the VLM enables the mapping to extend to data from other domains, as demonstrated in our experiments and in (Mazzaglia et al., 2024). This enables multi-task behavior learning by mapping diverse external VLM embeddings $e_g$ into corresponding WM goal states $z_g$:

$$z_g \sim Q_\psi(\cdot \mid e_g) \quad (6)$$

### 4.3. Behavior Learning

When a task prompt is provided, the agent performs standard GCRL within the world model to reach the goal state $z_g$ mapped from the prompt. The key challenge then becomes determining the reward function used to guide learning towards the goal. Previous methods typically employ cosine similarity with the goal state as the reward function (Hafner et al., 2022; Mazzaglia et al., 2024). However, we observe that this approach often fails due to reward hacking, particularly in locomotion tasks (Section 5.1). To address this, we propose using temporal distance (Mendonca et al., 2021) as an alternative reward signal.

**Temporal Distance Predictor.** We introduce a model $D_\theta$ to predict the temporal distance between two WM states. The temporal distance represents the number of timesteps required to transition from one state to another in the WM, effectively capturing the agent's progression towards the goal state when used in the reward function. We sample states pairs $z_t, z_{t+c}$ from the same trajectory sequence in the offline dataset, encoded by the pre-trained WM, and feed these pairs into $D_\theta$ to predict the distance $c$, normalized by the sequence length $T$. Also, we incorporate negative sample pairs $z^i, z^j$ from different trajectories, where the target for these pairs is the maximum possible distance in terms of timesteps:

$$\begin{aligned} \min_{D_\theta} \quad & \text{MSE}\left( D_\theta(z_t, z_{t+c}), \frac{c}{T} \right) \\ \min_{D_\theta} \quad & \text{MSE}\left( D_\theta(z^i, z^j), 1 \right) \end{aligned} \quad (7)$$

The insights behind these loss designs are widely adopted in prior work (Mendonca et al., 2021; Hartikainen et al., 2019). Temporal distance provides a more robust reward signal by extracting environmental dynamic information, promoting goal-oriented behavior and capturing deeper task semantics beyond visual details. The temporal distance predictor is also pre-trained before policy learning.

**Goal-conditioned Policy Learning.** The behavior learning process in the WM becomes straightforward. After imagining trajectory rollouts in the WM, we employ the learned $D$ to predict the temporal distance between each state and the mapped goal state, assigning rewards to each timestep in the imagined trajectories:

$$r_D(z_t, z_g) = -D_\theta(z_t, z_g) \quad (8)$$

Using this reward signal, we apply Dreamer-style Actor-Critic learning to train a goal-conditioned policy and value

*Table 1.* Normalized test performance of FOUNDER and baselines on DMC and Kitchen benchmarks. Mean scores (higher is better) with standard deviation are recorded across 6 seeds for each task.

| Task | GenRL | WM-CLIP | GenRL-TempD | FOUNDER w/o TempD | FOUNDER |
|---|---|---|---|---|---|
| Cheetah Stand | $0.93 \pm 0.03$ | $0.93 \pm 0.03$ | $0.42 \pm 0.04$ | $0.91 \pm 0.01$ | $\mathbf{1.02 \pm 0.01}$ |
| Cheetah Run | $0.68 \pm 0.06$ | $0.51 \pm 0.04$ | $0.37 \pm 0.06$ | $0.21 \pm 0.01$ | $\mathbf{0.81 \pm 0.02}$ |
| Cheetah Flip | $-0.04 \pm 0.01$ | $-0.11 \pm 0.11$ | $0.06 \pm 0.05$ | $-0.26 \pm 0.01$ | $\mathbf{0.97 \pm 0.02}$ |
| Walker Stand | $0.81 \pm 0.16$ | $\mathbf{1.01 \pm 0.02}$ | $0.92 \pm 0.06$ | $1.02 \pm 0.01$ | $1.01 \pm 0.02$ |
| Walker Walk | $\mathbf{0.95 \pm 0.02}$ | $\mathbf{0.95 \pm 0.03}$ | $0.42 \pm 0.10$ | $0.19 \pm 0.02$ | $0.94 \pm 0.04$ |
| Walker Run | $\mathbf{0.81 \pm 0.02}$ | $0.69 \pm 0.05$ | $0.68 \pm 0.03$ | $0.21 \pm 0.01$ | $0.78 \pm 0.04$ |
| Walker Flip | $0.48 \pm 0.04$ | $\mathbf{0.59 \pm 0.04}$ | $0.50 \pm 0.04$ | $0.28 \pm 0.02$ | $0.47 \pm 0.03$ |
| Stickman Stand | $0.60 \pm 0.11$ | $0.41 \pm 0.06$ | $0.49 \pm 0.05$ | $0.53 \pm 0.04$ | $\mathbf{0.91 \pm 0.04}$ |
| Stickman Walk | $0.83 \pm 0.03$ | $0.69 \pm 0.13$ | $0.84 \pm 0.07$ | $0.26 \pm 0.03$ | $\mathbf{0.91 \pm 0.03}$ |
| Stickman Run | $0.38 \pm 0.03$ | $0.37 \pm 0.04$ | $0.38 \pm 0.03$ | $0.17 \pm 0.00$ | $\mathbf{0.48 \pm 0.02}$ |
| Stickman Flip | $0.29 \pm 0.05$ | $\mathbf{0.62 \pm 0.03}$ | $0.38 \pm 0.04$ | $0.25 \pm 0.03$ | $0.41 \pm 0.03$ |
| Quadruped Stand | $0.95 \pm 0.06$ | $0.84 \pm 0.20$ | $\mathbf{0.97 \pm 0.04}$ | $\mathbf{0.99 \pm 0.02}$ | $\mathbf{0.98 \pm 0.01}$ |
| Quadruped Walk | $0.73 \pm 0.19$ | $0.64 \pm 0.21$ | $0.60 \pm 0.17$ | $0.51 \pm 0.02$ | $\mathbf{0.90 \pm 0.05}$ |
| Quadruped Run | $0.72 \pm 0.21$ | $0.58 \pm 0.13$ | $0.51 \pm 0.09$ | $0.52 \pm 0.01$ | $\mathbf{0.94 \pm 0.03}$ |
| Kitchen Light | $0.00 \pm 0.00$ | $0.35 \pm 0.48$ | $0.92 \pm 0.28$ | $\mathbf{1.00 \pm 0.00}$ | $0.97 \pm 0.18$ |
| Kitchen Slide | $0.62 \pm 0.49$ | $\mathbf{1.00 \pm 0.00}$ | $\mathbf{1.00 \pm 0.00}$ | $0.97 \pm 0.18$ | $\mathbf{1.00 \pm 0.00}$ |
| Kitchen Microwave | $\mathbf{1.00 \pm 0.00}$ | $0.63 \pm 0.48$ | $\mathbf{1.00 \pm 0.00}$ | $0.98 \pm 0.13$ | $\mathbf{1.00 \pm 0.00}$ |
| Kitchen Burner | $0.35 \pm 0.48$ | $0.10 \pm 0.30$ | $0.63 \pm 0.48$ | $\mathbf{1.00 \pm 0.00}$ | $0.60 \pm 0.49$ |
| Kitchen Kettle | $\mathbf{0.35 \pm 0.48}$ | $0.07 \pm 0.25$ | $0.05 \pm 0.22$ | $0.05 \pm 0.22$ | $\mathbf{0.33 \pm 0.47}$ |
| Overall | 0.60 | 0.57 | 0.59 | 0.52 | **0.81** |

function:

Action model: $a_t \sim \pi\left(a_t \mid z_t, z_g\right)$

Value model: $v\left(z_t, z_g\right) \approx \mathbb{E}_{\pi, \mathrm{WM}}\left[\sum_{k=t}^{H} \gamma^{k-t} r_D\left(z_t, z_g\right)\right]$ (9)

The behavior learning phase of FOUNDER integrates the high-level generalizable knowledge encoded in the VLM, the environment modeling capability of the world model, the task grounding ability of the mapping function, and the reward signal generation of the temporal distance predictor. The task-agnostic nature of these components allows for their potential application to a wide range of downstream tasks, facilitating open-ended multi-modal task specification and solving in embodied environments.

## 5. Experiments

We empirically evaluate the performance of FOUNDER in solving open-ended tasks specified by text or videos, using four locomotion environments (*Cheetah*, *Walker*, *Quadruped* and *Stickman* proposed in (Mazzaglia et al., 2024)) from the DeepMind Control Suite (DMC) (Tassa et al., 2018), and one manipulation environment (Franka Kitchen (Gupta et al., 2019)). For video tasks, we place special focus on cross-domain videos that cause significant visual gap. Furthermore, we validate the reliability of the reward function learned by FOUNDER. Finally, we extend FOUNDER to Minecraft, a more complex environments with challenging task instructions and more intricate observa-

tions. Details of environments and tasks are in Appendix A.

**Experimental Setup.** The agent is required to solve diverse downstream multi-modal tasks using a reward-free offline dataset and a Video-Language Model. We collect offline data consisting of image-action pairs for each domain, which are generated through exploration strategies like Plan2Explore (Sekar et al., 2020), and from the training buffers of RL agents designed to complete particular tasks. The agent must generate necessary learning signals for the underlying task and learn a corresponding policy. Details of our pre-collected data can be found in Appendix B. To ensure a fair comparison, FOUNDER and all baseline methods employ the same VLM, InternVideo2 (Wang et al., 2025), as adopted by (Mazzaglia et al., 2024).

**Baselines.** Model-free offline RL methods like IQL (Kostrikov et al., 2021b) and TD3+BC (Fujimoto & Gu, 2021) involve the cumbersome process of retraining from scratch for every task, and zero-shot FM-generated rewards using task prompts and observations has been shown to perform poor in (Mazzaglia et al., 2024). Therefore, we compare FOUNDER mainly with model-based methods. We select **GenRL**, which performs step-by-step state sequence matching in learning mapping and policy, **WM-CLIP** (Mazzaglia et al., 2024), which learns a mapping from WM states to FM representations and uses the similarity in FM space as the reward. We also evaluate **FOUNDER w/o TempD**, which uses cosine similarity in reward function during behavior learning. The final baseline is **GenRL-TempD**, which replaces the distance metric in GenRL with tempo-

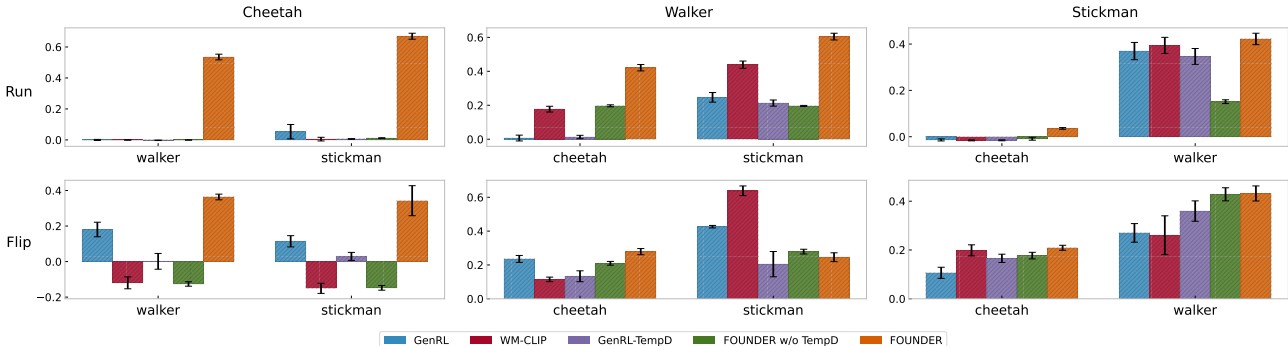

*Figure 2.* Normalized evaluation performance on cross-embodiment tasks built upon DMC. Each row corresponds to one of the *Run* or *Flip* tasks, while each column represents the domain in which the agent is evaluated. Each subplot presents the results of respectively using videos from the remaining two domains as task prompts. This yields 6 domain combinations: (Cheetah, Walker), (Cheetah, Stickman), (Walker, Cheetah), (Walker, Stickman), (Stickman, Cheetah), and (Stickman, Walker), each evaluated on *Run* or *Flip*, totaling 12 evaluation tasks. Mean scores with standard deviation are recorded across 4 seeds.

ral distance, designed to assess whether temporal distance alone contributes to the performance of FOUNDER. Each method pre-trains WM and other components for 500K gradient steps, and then is constrained to a maximum of 50K gradient updates in behavior learning. Implementation details of these model-based baselines are in Appendix C. The comparison of our approach with model-free methods, including an analysis of efficiency and computation costs, is detailed in Appendix D.1 and Appendix D.2.

### 5.1. Language Task Solving on DMC and Kitchen

We evaluate the ability of interpreting and solving open-ended text-based tasks of FOUNDER on DMC and Franka Kitchen. For DMC, we select four locomotion tasks that are generic across different agent embodiments: *Stand*, *Walk*, *Run* and *Flip*, and evaluate all possible environment-task combinations. For Kitchen, we evaluate on five commonly used tasks. We incorporate the aligner from GenRL to denoise the text embeddings of task prompts into visual embeddings. The results are presented in Table 1, where performance is normalized between the expert policy and a random policy.

FOUNDER achieves the best overall performance, ranking among the highest in 14 out of 19 tasks. GenRL struggles on kitchen tasks with inherently static properties, underscoring its lack of temporal awareness and the timestep disalignment introduced by step-by-step trajectory matching. The poor performance of WM-CLIP emphasizes the importance of aligning representations in WM state space rather than in VLM space, as WM states encapsulate actionable information beyond surface-level visual or linguistic semantics. GenRL-TempD does not outperform GenRL, indicating that temporal distance alone does not contribute significantly to performance; its effectiveness emerges only when integrated within the GCRL paradigm of FOUNDER,

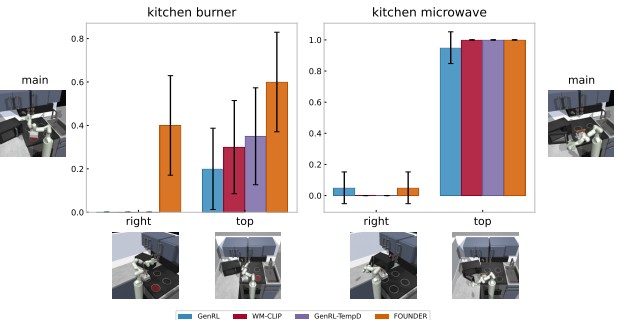

*Figure 3.* Performance of FOUNDER and baselines over 4 seeds on two tasks in Kitchen. The agent's observations is captured from the Main viewpoint, while task video prompts are provided from the Right or Top viewpoints, yielding 4 cross-view tasks.

as is demonstrated by the low performance of FOUNDER w/o TempD.

Moreover, when studying the failure cases of GenRL and FOUNDER w/o TempD, we find that their failures can be attributed to reward hacking which leads to unintended outcomes. In these scenarios, GenRL or FOUNDER w/o TempD agents only mimic the visual features of the task without truly accomplishing it, revealing their inability to interpret and exploit deep-level task semantics. A GenRL agent might simply wave its arms near the light switch instead of actually turning it on. A FOUNDER agent tasked with running may remain stationary if not use the temporal distance predictor. We provide a case study for these failures on our website https://sites.google.com/view/founder-rl, and in Appendix E.3.

### 5.2. Cross-domain Video Task Solving

We then evaluate FOUNDER's capability in solving video-based tasks, with a particular focus on cross-domain scenar-

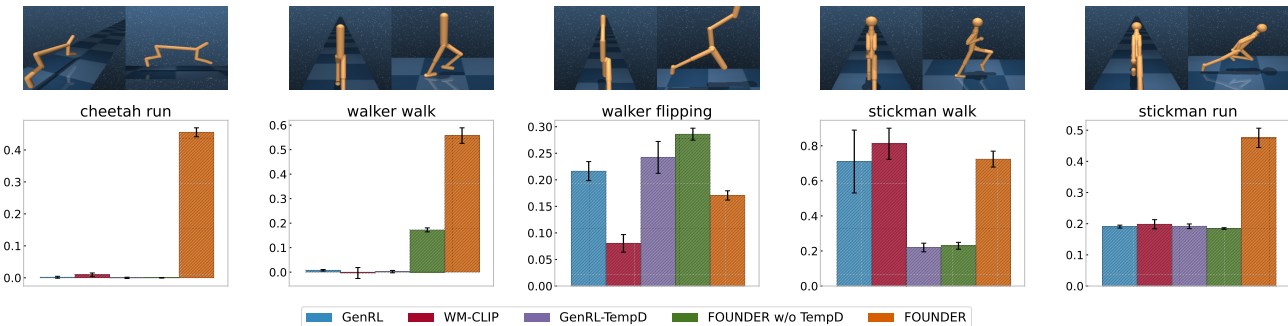

Figure 4. Performance evaluation of FOUNDER and baselines over 4 seeds across 5 cross-viewpoint video tasks in DMC. Visualizations of video prompts indicating the task semantics and the agent's observation in the corresponding environment is presented at first row.

ios where the visual appearance of task videos differs significantly from the agent's environment observations. This includes videos captured from alternative camera viewpoints or showcasing behaviors performed by morphologically different agent embodiments. Notably, each task is defined by a single 8-frame video, as InternVideo2 can process a maximum of 8 frames. This constraint further increases the difficulty of the tasks, resulting in a problem setting of Cross-Domain (Third-Person) One-Shot Visual Imitation Learning (Stadie et al., 2017; Wan et al., 2023; Huang et al., 2024), which is challenging for prior methods.

For cross-embodiment evaluation, we assess behavior learning for the *Run* and *Flip* tasks within one of the three domains—*Cheetah*, *Walker*, or *Stickman*—while using task videos sourced from the remaining two domains. This setup results in six domain combinations and a total of 12 tasks. The evaluation results are presented in Figure 2, where FOUNDER achieves the best performance on 11 out of 12 tasks. Notably, FOUNDER is the only method that succeeds in Cheetah-based domains, with other baselines completely failing to connect tasks specified in *Walker* and *Stickman* videos into *Cheetah* embody. FOUNDER excels in challenging domain transfers, such as (*Cheetah*, X) or (X, *Cheetah*), where the morphological differences between *Cheetah* and the other two domains are significantly larger than those between *Walker* and *Stickman*.

In the cross-view evaluation, we select several tasks from DMC and Kitchen where the task video prompts are captured from alternative camera viewpoints. The evaluation results are presented in Figure 3 and Figure 4. FOUNDER outperforms or matches the performance of baselines across most tasks, and is the only method to learn a successful policy in *Cheetah Run*, *Walker Walk* and *Kitchen Burner* (Right). Notably, GenRL exhibits the poorest performance among all baselines in Kitchen tasks, where the domain gap across different viewpoints is huge.

The above results demonstrate that FOUNDER captures deep-level task semantics beyond visual features and ac-

Table 2. Evaluation of the consistency between learned pseudo rewards and ground-truth rewards, averaged on 7 tasks in DMC. The results for each task are shown in Appendix D.4.

| Methods | Corr↑ | Regret↓ | Precision↑ | Recall↑ | F1↑ |
|---|---|---|---|---|---|
| GenRL | 0.12 | 0.37 | 0.47 | 0.44 | 0.44 |
| WM-CLIP | 0.40 | 0.26 | 0.61 | **0.69** | **0.63** |
| GenRL-TempD | 0.05 | 0.75 | 0.46 | 0.46 | 0.40 |
| FOUNDER w/o TempD | -0.02 | 0.90 | 0.16 | 0.15 | 0.15 |
| FOUNDER | **0.54** | **0.07** | **1.0** | 0.47 | 0.59 |

curately aligns them with the corresponding physical states in the world model. While (Mazzaglia et al., 2024) shows that GenRL can effectively translate real-world videos into videos in the simulator and learn a policy, our experiments reveal its failure to interpret the semantics of cross-domain, simulation-generated videos. This contrast highlights that GenRL primarily operates as a style-transfer-like approach, aligning visual appearances rather than capturing deeper physical states. In contrast, FOUNDER excels in extracting and grounding underlying physical information from task videos to WM space, making it more effective in scenarios requiring true semantic understanding.

Furthermore, to provide a more comprehensive evaluation of FOUNDER's performance on out-of-distribution video task prompts, we also conduct experiments on real-world video tasks, following the protocol in (Mazzaglia et al., 2024), as detailed in Appendix D.3. The results again indicate that FOUNDER demonstrates solid performance in comparison to GenRL.

### 5.3. Evaluation of the Learned Reward Function

We assess the learned reward functions of FOUNDER and baseline methods in comparison with ground-truth rewards. The learned reward functions depend on the pre-trained WM state space, the mapping function, and the similarity metric, comprehensively reflecting the properties of each method. We use the metrics of **RankCorr** and **Regret@1** to evaluate whether the learned pseudo-reward function ac-

*Figure 5.* Performance of FOUNDER and baselines over 3 seeds across 5 tasks in Minecraft during behavior learning. Each task is specified in a text prompt. The solid curves and the shaded region indicate the average episodic success rates and the 95% confidence intervals across different runs. We apply a moving average to smooth the curves.

curately estimates the ground-truth reward. Additionally, we adopt the evaluation approach from (Fan et al., 2022), which converts the reward model into a binary success classifier, and measure performance using **Precision**, **Recall**, and **F1**-score. We evaluate all methods on the testing buffers stored during language task behavior learning (Section 5.1). The metrics are computed at the trajectory level, where the ground-truth and pseudo rewards for each timestep are averaged over each trajectory. More detailed information about the evaluation and metrics can be found in Appendix C.4.

The results are presented in Table 2. We observe that FOUNDER achieves the best consistency with the ground-truth reward as measured by Correlation and Regret. For binary classification metrics, FOUNDER demonstrates perfect Precision, though it is outperformed by WM-CLIP on Recall and the final F1 score. However, during policy learning based on our assigned pseudo-rewards, avoiding the misclassification of low-reward behavior as high-reward is far more critical than identifying all high-reward behaviors. This mirrors cost-sensitive learning (Zhou & Liu, 2005), where the cost of False Positive samples is much larger than that of False Negative samples. In this context, precision outweighs recall, as mistakenly favoring low-reward behaviors can lead to reward hacking and undesirable outcomes, as seen with FOUNDER w/o TempD. Overall, FOUNDER exhibits the highest consistency in reward estimation, effectively translating task prompts into WM goal states and generating reliable reward functions that guide policy learning.

### 5.4. Extending FOUNDER to Minecraft Environment

We then try extending FOUNDER to more challenging open-ended embodied environments such as Minecraft, where the visual observations are significantly more complex and the task instructions are harder to interpret and complete. We adopt the Minedojo benchmark (Fan et al., 2022), which introduces thousands of diverse tasks specified in free-form language. Specifically, we evaluate FOUNDER and baselines on five tasks: "Hunt Cow", "Shear Sheep", "Chop Trees", "Milk Cow", and "Hunt Sheep" within different

biomes, which are specified in text prompts, listed in Table 3. The agent relies solely on visual observations for decision-making, without access to manually designed dense rewards, success signals, or underlying state information.

We evaluate the performance of FOUNDER on Minecraft tasks with GenRL and MineCLIP-IQL. Following the same setup as previous experiments, the agents are provided only with a VLM and an offline dataset for task understanding and adaptation. For both FOUNDER and GenRL, we first perform pre-training on the offline dataset for 500K gradient steps and then conduct behavior learning for downstream tasks for 100K steps. Additionally, we incorporate MineCLIP in (Fan et al., 2022), a text-conditioned reward model pre-trained on an internet-scale Minecraft knowledge base, to establish a model-free offline RL baseline (IQL), where MineCLIP is used to assign rewards to the offline dataset for each text task. The resulting MineCLIP-IQL, is an oracle method for comparison. Implementation details of MineCLIP-IQL can be found in Appendix C.3. The evaluation results are presented in Figure 5.

We observe that FOUNDER method achieves similar performance to GenRL on two of the five tasks while consistently surpassing GenRL on the other three. Notably, FOUNDER's average success rate on these three tasks consistently exceeds GenRL's upper confidence interval, demonstrating statistically significant gains.

Moreover, FOUNDER matches or surpasses the oracle MineCLIP-based approach in 3 out of 5 tasks, underscoring the effectiveness of our paradigm for leveraging FMs in RL. Instead of finetuning FMs with vast amounts of related Internet data, our approach learns an extra module that connects FMs to WMs using limited environment data, providing a more efficient and adaptable solution.

## 6. Discussion

We introduce FOUNDER, a framework that integrates FMs and WMs to enable open-ended task specification and completion in embodied environments. By grounding FM-

generated task representations into corresponding goal states within the WM, `FOUNDER` effectively bridges high-level knowledge with low-level dynamics, facilitating efficient GCRL for diverse downstream tasks. The framework's physical-state-aligned mapping function, paired with a temporal-distance-based reward signal, allows the agent to go beyond visual features, obtain a deeper interpretation of the task, and rapidly adapt to multi-modal prompts. `FOUNDER` demonstrates superior performance in complex or cross-domain environments where prior methods struggle, underscoring its potential for robust and generalizable embodied task solving.

Our method relies on offline environment data to learn the WM and mapping function, meaning that prompt elements not represented in the dataset may not be properly grounded. As the performance of `FOUNDER` is upper-bounded by the offline data, future work should explore more data collection strategies or incorporate real-world videos into WM learning. Additionally, while our current experiments mainly focus on short-horizon tasks, integrating the reasoning capabilities of FMs into our framework could enhance its ability to handle long-horizon tasks that require task decomposition. Incorporating more powerful sequence modeling methods could also improve long-term memory within the agent to handle complex, long-horizon tasks. We leave these explorations to future work.

## Acknowledgements

This work is partially supported by National Science and Technology Major Project (2022ZD0114805), NSFC (No. 62476123), Postgraduate Research & Practice Innovation Program of Jiangsu Province (KYCX25_0326), and the Young Scientists Fund of the National Natural Science Foundation of China (PhD Candidate) (624B200197).

## Impact Statement

This paper aims to advance the field of machine learning by proposing a framework that integrates the power of FMs and WMs to facilitate open-ended embodied task solving across domains such as gaming, motion control, and autonomous manipulation. This has the potential to generate significant positive impacts on industry and society.

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

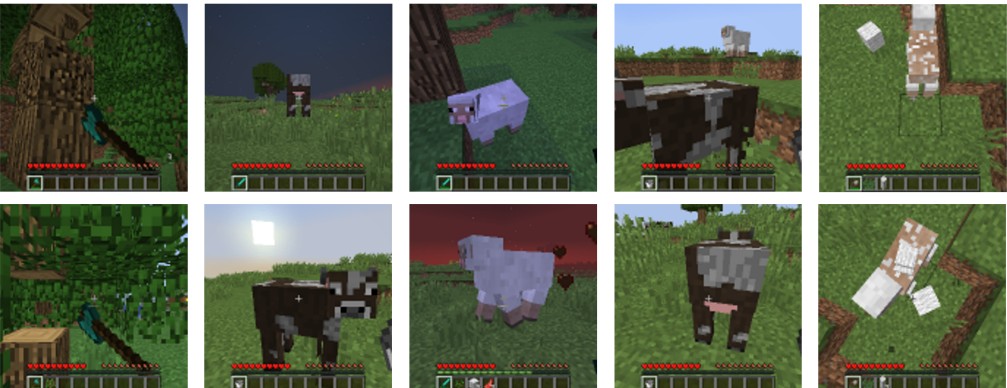

*Figure 6.* Demonstration of the Minecraft environment, where the agent performs tasks such as chopping trees, hunting cows, milking cows, hunting sheep, and shearing sheep.

## A. Environments and Tasks

**Minecraft environment.** The Minecraft environment is based on a popular sandbox game and is a widely recognized benchmark for AI research due to its complex and dynamic nature. The raw observation space consists of images with a resolution of 160×256. Minedojo has an 18-dimensional multi-discrete action space, allowing for diverse interactions such as movement, tool usage, and resource collection. However, since the original action space is too large and we only evaluate on task categories of "Harvest" and "Combat", we modify the action space in the same way as (Ma et al., 2023; Zhou et al., 2024), resulting in a 5-dimensional multi-discrete action space. We select 5 programmatic tasks (Fan et al., 2022) including "Hunt Cow", "Shear Sheep", "Chop Trees", "Milk Cow", and "Hunt Sheep". The prompts we use for each task are the same as Minedojo, which are listed in Table 3. A demonstration of the environment and tasks is shown in Figure 6. In our experiment, the resolution of image observations is 128×128 and the action repeat is 1. The evaluation metric is the success rate that can better reflect the agent's performance compare to manually-designed rewards.

**DMC and Franka Kitchen** We use the implementation in (Mazzaglia et al., 2024) of tasks not originally existing in the DMC benchmark, such as several "stand" and "flip" tasks, for evaluation. The text prompts of language-based tasks Section 5.1 evaluated in DMC and Franka Kitchen are listed in Table 3. The text prompts we use in FOUNDER for DMC and Kitchen are mostly similar to GenRL (Mazzaglia et al., 2024). When evaluating baseline methods based on GenRL, we remain using the original prompts in the GenRL paper. In our experiment, the resolution of image observations is 64×64 and the action repeat is 2.

## B. Offline Dataset

We reuse the offline dataset collected by (Mazzaglia et al., 2024) for DMC and Kitchen, which consists of trajectories generated through performing Plan2Explore in the environment (Sekar et al., 2020) and from training buffers of RL agents designed to complete particular tasks within the corresponding domain. Additionally, (Mazzaglia et al., 2024) does not provide the Quadruped dataset, so we collect it on our own. For Minecraft, we use the same strategy to collect the environment data in Minedojo. Details of the offline dataset information are listed in Table 4.

## C. Implementation Details

### C.1. Implementation Details of FOUNDER

**Pretraining.** At first, we train a DreamerV3 world model (WM) where WM states include deterministic and stochastic ones. The stochastic states are sampled from the categorical distribution as in DreamerV3. We then fix the WM to learn the remaining components (mapping function and temporal distance predictor), by encoding offline trajectory data to WM state space. During mapping function learning, the states mapped by $Q_\psi$ are in the same space of WM states, also include deterministic and stochastic parts. Therefore, We parameterize the two parts in the output layer of $Q_\psi$ for computing the KL divergence in distribution alignment, and respectively using Gaussian and categorical distribution to parameterize the

*Table 3.* Prompts for each task in language-based evaluation, as well as expert and random agent score

| Task | Prompt | Specialized agent score | Random agent score |
|---|---|---|---|
| cheetah stand | standing on the floor like a human | 930 | 5 |
| cheetah run | running like a quadruped | 890 | 9 |
| cheetah flip | quadruped rotating flips | 990 | 250 |
| walker stand | standing up straight like a human | 970 | 150 |
| walker walk | walk fast clean like a human | 960 | 45 |
| walker run | run fast clean like a human | 770 | 30 |
| walker flip | doing backflips | 720 | 20 |
| stickman stand | standing like a human | 970 | 70 |
| stickman walk | robot walk fast clean like a human | 960 | 35 |
| stickman run | robot run fast clean like a human | 830 | 25 |
| stickman flip | doing flips like a human | 790 | 45 |
| quadruped stand | spider standing | 990 | 15 |
| quadruped run | running fast clean like a quadruped spider | 930 | 10 |
| quadruped walk | walking fast clean like a quadruped spider | 960 | 10 |
| kitchen light | activate the light | 1 | 0 |
| kitchen slide | slide cabinet above the knobs | 1 | 0 |
| kitchen microwave | opening the microwave fully open | 1 | 0 |
| kitchen burner | the burner becomes red | 1 | 0 |
| kitchen kettle | pushing up the kettle | 1 | 0 |
| minecraft hunt cow | hunt a cow | / | / |
| minecraft shear sheep | shear a sheep | / | / |
| minecraft chop trees | chop trees to obtain log | / | / |
| minecraft milk cow | obtain milk from a cow | / | / |
| minecraft hunt sheep | hunt a sheep | / | / |

distribution of the two parts. Additionally, we learn an ensemble of mapping function, where we learn multiple output layers of stochastic state. The learning procedure of this ensemble module is similar to plan2explore (Sekar et al., 2020), and the ensemble number is 5. During our learning of the temporal distance predictor $D_\theta$, we use the two-hot trick in $D_\theta$'s output layer, which is also used in the output layer of the reward model in DreamerV3.

**Behavior learning.** We use the mean value of the goal distribution mapped from the VLM embedding as the goal state in DMC and Kitchen, while sampling goal states from the goal distribution in Minecraft for exploring more behaviors. FOUNDER and all the baseline methods do not use a MOPO-style reward punishment (Yu et al., 2020; Lu et al., 2022; Wan et al., 2024) for unreliable transition prediction during policy learning, same as the GenRL code (Mazzaglia et al., 2024), ensuring a fair comparison. We apply normalization on the temporal-distance-based rewards for efficient policy learning.

**Policy rollout.** When acting with the environment, FOUNDER's goal-conditioned policy needs a goal state for every timestep's decision making. We use the same goal sampling approach as in behavior learning, while only selecting the goal every 8 steps. This ensures a short-period decision-making consistency and robustness (Hafner et al., 2022).

### C.2. Implementation Details of baselines in DMC and Kitchen

For GenRL, we follow the code in (Mazzaglia et al., 2024) to train the MFWM learn policies for different tasks. For WM-CLIP, we follow the instruction in (Mazzaglia et al., 2024) to train a reversed mapping from WM state space to VLM space, minimizing the MSE loss of the predicted VLM embedding with the ground-truth one. For GenRL-TempD, we use the trained WM in GenRL to learn a temporal distance predictor, using only stochastic states as WM states input to the predictor since GenRL's WM encoder is only a visual tokenizier (Mazzaglia et al., 2024) and does not include deterministic states during trajectory encoding. Then we use each step's predicted temporal distance between imagined trajectories and connected task sequence as the reward in behavior learning, replacing the original cosine similarity.

Table 4. Datasets composition.

| Domain | num of observations | Subset | Subcount |
|---|---|---|---|
| **cheetah** | 1.8M | cheetah run | 820K |
| | | cheetah plan2explore | 1M |
| **walker** | 2.5M | walker stand | 500k |
| | | walker walk | 500k |
| | | walker run | 500k |
| | | walker plan2explore | 1M |
| **stickman** | 2.5M | stickman stand | 500k |
| | | stickman walk | 500k |
| | | stickman run | 500k |
| | | stickman plan2explore | 1M |
| **quadruped** | 2.5M | quadruped stand | 500k |
| | | quadruped walk | 500k |
| | | quadruped run | 500k |
| | | quadruped plan2explore | 1M |
| **kitchen** | 3.6M | kitchen light | 700k |
| | | kitchen slide | 700k |
| | | kitchen microwave | 700k |
| | | kitchen bottom burner | 700k |
| | | kitchen plan2explore | 800k |
| **minecraft** | 3.4M | minecraft hunt cow | 1M |
| | | minecraft shear sheep | 610K |
| | | minecraft chop trees | 810K |
| | | minecraft plan2explore | 1M |

## C.3. Implementation of MineCLIP-IQL

We employ MineCLIP-IQL as a strong baseline for our experiments in the Minecraft environment. For policy learning, we utilize Implicit Q-Learning (IQL) (Kostrikov et al., 2021a), implemented via OfflineRL-Lib (Gao & Kong, 2023).

For the reward function, we derive the MineCLIP score by measuring the similarity between the video embedding and the prompt embedding, using the direct reward approach from (Fan et al., 2022). Specifically, the reward is computed as:

$$r = \max\left(P_G - \frac{1}{N_T}, 0\right) \tag{10}$$

where $N_T$ denotes the number of prompts passed to MineCLIP, and $\frac{1}{N_T}$ represents the baseline probability of randomly guessing the correct text-video match. To prevent highly uncertain probability estimates from falling below the random baseline, $r$ is thresholded at zero. Following the policy learning approach in (Fan et al., 2022), we perform IQL on 512-dimensional vectors, which are low-dimensional embeddings of the original image observations in the offline dataset, encoded by MineCLIP's pre-trained encoder.

## C.4. Evaluation Metrics

The metrics and methods used for evaluating the pseudo rewards generated by FOUNDER and baselines during behavior learning are defined and outlined as follows. Since we calculate all the metrics in Section 5.3 at the trajectory level, they measure the relationship between the pseudo-returns and the real returns of the trajectories.

**Rank correlation** quantifies the correlation between the ordinal rankings of the generated pseudo returns $\hat{V}_{1:N}$ and the corresponding ground-truth ones $V_{1:N}$. It is formulated as:

$$\text{RankCorr} = \frac{\text{Cov}(V_{1:N}, \hat{V}_{1:N})}{\sigma(V_{1:N})\sigma(\hat{V}_{1:N})}, \tag{11}$$

where $1 : N$ represents the indices of the trajectories.

**Regret@k** measures the gap between the return of the best trajectory and the return of the best trajectory selected within the top-k set, where the top-k set is determined based on pseudo returns. It is expressed as:

$$\text{Regret@k} = \max_{i \in 1:N} V_i^\pi - \max_{j \in \text{topk}(1:N)} V_j^\pi, \tag{12}$$

where $\text{topk}(1 : N)$ denotes the indices of the top $K$ trajectories, ranked according to their pseudo returns $\hat{V}$.

**Binary classification-based evaluation** Following (Fan et al., 2022), we convert our trained reward model into a binary classifier and evaluate its performance. (Fan et al., 2022) incorporates human expert to label 100 good and 100 bad trajectories as ground-truth labels, and then use K-means clustering (K=2), to determine a decision boundary $\delta$ from the centroids' mean, classifying trajectories above $\delta$ as successful. Following their practice, we use all methods' testing buffers stored during language task behavior learning (Section 4.3) and use the ground-truth reward to label the ground-truth "good" or "bad" trajectories. We then compute step-wise pseudo rewards using FOUNDER and baseline methods, averaging them per trajectory. We then use the same clustering-based approach to find the decision boundary and obtain the classification results with ground-truth labels. The classifier's performance is measured using Precision, Recall, and F1-score, showing high agreement with ground-truth labels, making it a reliable automatic evaluation method.

## D. Additional Results

### D.1. Model-Free Baselines

Prior work (Mazzaglia et al., 2024) has proven that traditional model-free offline RL methods (IQL, TD3+BC, TD3) perform poorly in our multi-task reward-free setting compared to model-based counterparts, and we mainly compare FOUNDER to model-based baselines in Table 1. Traditional single-task model-free methods require training from scratch for each task, making them inefficient and impractical for multi-task solving.

To ensure a more comprehensive evaluation, we include HILP (Park et al., 2024), a multi-task model-free method that achieves strong performance both in zero-shot offline RL and goal-conditioned RL, and TD3, the best-performing single-task model-free baseline according to GenRL's experiment (Mazzaglia et al., 2024). We assess their performance across eight tasks in the Cheetah and Kitchen domains, and the results are presented in Table 5. Following (Mazzaglia et al., 2024) for implementing single-task model-free methods like TD3, we use a learnable DrQ-v2 encoder to process the visual observations and a frame stack of 3. The reward at each timestep is calculated as the cosine similarity between the VLM embedding of the text task prompt and the video embedding of the last k frames (k=8, same as FOUNDER and GenRL). For the implementation of HILP, we leverage their official code for zero-shot offline RL, and adopt the VLM-produced reward same as in TD3, for HILP's downstream zero-shot policy learning.

*Table 5.* Normalized test performance of FOUNDER and model-free baselines on Cheetah and Kitchen benchmark. We also include the performance of GenRL for a clearer comparison.

| Task | TD3 | HILP | GenRL | FOUNDER |
|---|---|---|---|---|
| Cheetah Stand | $0.71 \pm 0.11$ | $0.56 \pm 0.24$ | $0.93 \pm 0.03$ | $\mathbf{1.02 \pm 0.01}$ |
| Cheetah Run | $0.28 \pm 0.03$ | $0.22 \pm 0.14$ | $0.68 \pm 0.06$ | $\mathbf{0.81 \pm 0.02}$ |
| Cheetah Flip | $-0.03 \pm 0.03$ | $0.20 \pm 0.03$ | $-0.04 \pm 0.01$ | $\mathbf{0.97 \pm 0.02}$ |
| Kitchen Light | $0.10 \pm 0.18$ | $0.00 \pm 0.00$ | $0.00 \pm 0.00$ | $\mathbf{0.97 \pm 0.18}$ |
| Kitchen Microwave | $0.10 \pm 0.15$ | $0.00 \pm 0.00$ | $\mathbf{1.00 \pm 0.00}$ | $\mathbf{1.00 \pm 0.00}$ |
| Kitchen Slide | $0.72 \pm 0.27$ | $0.73 \pm 0.09$ | $0.62 \pm 0.49$ | $\mathbf{1.00 \pm 0.00}$ |
| Kitchen Burner | $0.37 \pm 0.12$ | $0.00 \pm 0.00$ | $0.35 \pm 0.48$ | $\mathbf{0.60 \pm 0.49}$ |
| Kitchen Kettle | $0.03 \pm 0.07$ | $0.00 \pm 0.00$ | $\mathbf{0.35 \pm 0.48}$ | $0.33 \pm 0.47$ |

We observe that FOUNDER consistently outperforms both single-task and multi-task model-free baselines. Model-free methods rely on zero-shot pseudo-rewards generated by FMs through cosine similarity between FM representations of task prompts and observations. However, the lack of embodied grounding in these FM rewards restricts their effectiveness. Notably, although HILP demonstrates strong performance in multi-task reward-free offline RL settings in a zero-shot manner,

when ground-truth rewards are given during downstream policy learning (Park et al., 2024), its performance deteriorates significantly when relying on FM-generated rewards, underscoring the importance of the grounding process. In contrast, model-based methods learn a world model (WM) that captures environment dynamics, facilitating knowledge reuse across diverse downstream tasks and enabling explicit grounding of FMs. This fundamental advantage motivates us to integrate WMs and FMs to facilitate open-ended task solving.

### D.2. Efficiency and Computational Costs

All experiments were conducted on an RTX 3090 GPU. For pretraining, training the MFWM in GenRL for 500k gradient steps requires approximately five days, whereas FOUNDER reduces this overhead to about three days by avoiding the seq2seq-style generative sequence modeling used in GenRL. Moreover, pretraining the Hilbert representation and foundation policy in HILP for 500k gradient steps takes around 30 hours.

For downstream tasks, training the actor-critic in the learned world model on a given task prompt for 50k gradient steps takes under five hours for both GenRL and FOUNDER. In the case of model-free methods, HILP enables zero-shot adaptation to downstream tasks, whereas single-task model-free methods like TD3 require approximately seven hours to train from scratch for 500k gradient steps.

We present a comprehensive comparison of the aforementioned methods—the single-task model-free TD3, the multi-task model-free HILP, GenRL, and FOUNDER —considering both performance and computational efficiency. Overall, we find that FOUNDER strikes a strong balance between performance and learning efficiency, as shown in Table 6.

*Table 6.* Overall comparison between TD3, HILP, GenRL and FOUNDER, considering both performance and computational efficiency

|  | TD3 | HILP | GenRL | FOUNDER |
|---|---|---|---|---|
| Performance | Low | Low | Medium | High |
| Multi-Task Adaptation | Learning from scratch ($\sim$ 7h) | Zero-shot | Finetuning ($\sim$ 5h) | Finetuning ($\sim$ 5h) |
| Pretraining Overhead | N/A | $\sim$ 30h | $\sim$ 120h | $\sim$ 72h |
| Overall Computational Cost | Low | Medium-Low | High | Medium |

### D.3. Real-world Video Tasks

We present experimental results on real-world video tasks using videos provided in GenRL's code repository. The used file names of real-world videos are 'person_standing_up_with_hands_up_seen_from_the_side', 'spider_draw', 'dog_running_seen_from_the_side', 'guy_walking', 'open_microwave' respectively in GenRL's code repository, for specifying each task of 'Cheetah Stand', 'Quadruped Walk', 'Cheetah Run', 'Stickman Walk' and 'Kitchen Microwave', respectively. The results are presented in Table 7.

*Table 7.* Normalized test performance of FOUNDER and GenRL on real-world video tasks. We also list the performance of GenRL and FOUNDER on corresponding language tasks as upper bounds for comparison.

| Task | GenRL_video | FOUNDER_video | GenRL_language | FOUNDER_language |
|---|---|---|---|---|
| Cheetah Stand | $1.05 \pm 0.00$ | $0.77 \pm 0.01$ | $0.93 \pm 0.03$ | $1.02 \pm 0.01$ |
| Quadruped Walk | $0.66 \pm 0.01$ | $0.99 \pm 0.01$ | $0.73 \pm 0.19$ | $0.90 \pm 0.05$ |
| Cheetah Run | $0.63 \pm 0.01$ | $0.60 \pm 0.01$ | $0.68 \pm 0.06$ | $0.81 \pm 0.02$ |
| Stickman Walk | $0.52 \pm 0.02$ | $0.85 \pm 0.02$ | $0.83 \pm 0.03$ | $0.91 \pm 0.03$ |
| Kitchen Microwave | $1.00 \pm 0.00$ | $1.00 \pm 0.00$ | $1.00 \pm 0.00$ | $1.00 \pm 0.00$ |
| Overall | $0.77 \pm 0.01$ | $0.84 \pm 0.01$ | $0.83 \pm 0.06$ | $0.93 \pm 0.02$ |

FOUNDER again demonstrates solid performance compared to GenRL when generalizing to real-world video task understanding and grounding, even matching GenRL's language-based task solving performance. These results, coupled with FOUNDER's performance on cross-embodiment and cross-viewpoint video tasks detailed in Section 5.2, confirm FOUNDER's strong generalization capabilities to out-of-distribution tasks.

## D.4. Reward Evaluation

We offer the complete reward evaluation results in Section 5.3 on each of the 7 tasks, as shown in Table 8.

# E. Additional Discussions

## E.1. Clarification on open-ended task solving

In our context, open-endedness specifically refers to agent's ability to ground and solve tasks specified through free-form user-defined task prompts via multimodal interfaces without predefined reward functions. FOUNDER develops this capability through integrating the high-level knowledge embedded in Foundation Models with the environment-specific modeling capabilities of World Models. The terminology "open-ended" is widely used in a similar way in (Fan et al., 2022; Team et al., 2021; Qin et al., 2024), which aim to build a generally capable agent, e.g. capable of solving free-form language tasks in (Fan et al., 2022).

## E.2. Is FOUNDER's performance gain solely comes from temporal-distance-based reward?

FOUNDER grounds VLM representations into corresponding WM states without GenRL's computationally intensive sequence matching and enables deeper task understanding. It is FOUNDER's overall architecture—not isolated reward engineering techniques like temporal distance—that drives the performance gains.

From Table 1, even without temporal distance, FOUNDER w/o TempD already outperforms GenRL on static tasks (Stand tasks and Kitchen tasks). This demonstrates the inherent efficacy of our grounding framework, independent of reward shaping. FOUNDER w/o TempD struggles in dynamic tasks (Walk, Run) since single-frame goal grounding loses temporal awareness compared to GenRL (which uses multi-step seq2seq alignment) when using cosine similarity as rewards. At this time, the role of TempD-based rewards is to provide temporal information to the agent, with a much lower training cost than GenRL's complex sequence matching. This is detailed in Appendix E.3.

Moreover, the performance degradation observed when simply adding temporal distance to GenRL (GenRL+TempD, Table 1) further suggests that, it is FOUNDER's overall architectural design, rather than isolated reward engineering, that drives the gains.

## E.3. Case study: the influence of temporal distance

As mentioned in Appendix E.2, temporal-distance-based rewards mainly serve to enhance temporal awareness and provide task-completion information for the FOUNDER agent. In this section, we present a study of the failure cases of FOUNDER w/o TempD on dynamic tasks. Analysis of GenRL's failure cases can be found on our website https://sites.google.com/view/founder-rl.

As shown in Table 1, while exhibiting strong performance on static tasks (Stand and Kitchen), FOUNDER w/o TempD struggles in dynamic tasks (Walk, Run). When we plot the return curves during policy learning of dynamic tasks, we observe that, the agent can perform well at the very beginning of the behavior learning stage, but its performance deteriorates as training progresses, eventually reaching very low performance by the end (Figure 7). We also discover that the real return and pseudo return curves exhibit completely opposite trends during behavior learning, as shown in Figure 7. The real performance worsens as the agent maximizes the pseudo reward, causing reward hacking problem.

We then visualize the resulting trajectories of the policy at the early stage and the final policy. Despite performing well at the beginning, we find that at the end, the agent's behavior becomes more static. The agent may stay at its initial position, appearing as if it is running or walking, but in reality, it is only 'running' or 'walking' at the original place without progressing forward, or moving forward at an extremely slow pace. This explains the poor final performance of FOUNDER w/o TempD in Table 1. Visualization of these trajectories can be found on our website https://sites.google.com/view/founder-rl.

We conclude that, reward functions based on cosine similarity or other direct distance metrics may lead the policy to mimic the visual appearance of the goal, while overlooking the underlying task semantics and multi-step movement, particularly in dynamic tasks like running or walking. Since FOUNDER-based method maps the target sequence to a single goal state in the world model, we hypothesize that using direct distance functions may result in a lack of temporal awareness.

At this time, the role of TempD-based rewards is to provide temporal information and crucial task-completion information

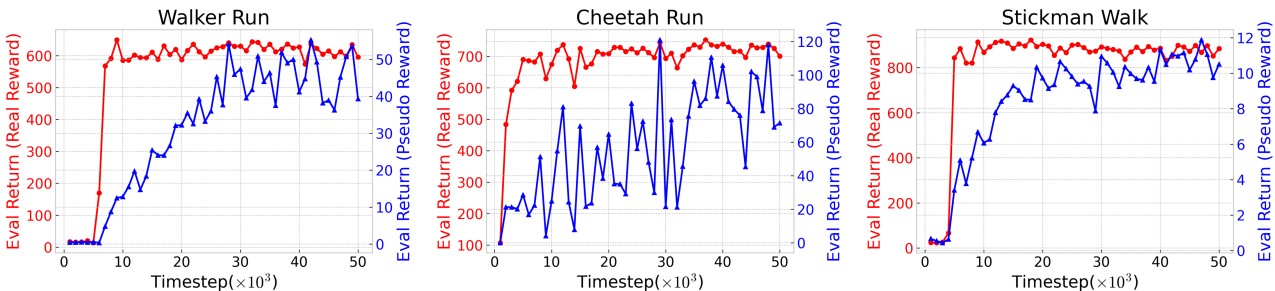

*Figure 7.* Testing returns of the FOUNDER w/o TempD agent during behavior learning stage for Walker Run, Cheetah Run and Stickman Walk in terms of real performance and pseudo return. As the agent maximizes pseudo-returns derived from cosine similarity (blue curves), the agent's performance (red curves) degrades over the course of training, starting strong (Point A) but ending poorly (Point B).

*Figure 8.* Testing returns of the FOUNDER agent during behavior learning stage for Walker Run, Cheetah Run and Stickman Walk in terms of real performance and pseudo return, when using TempD-based rewards. The trends of the two return curves are now generally consistent.

to the agent, with a much lower training cost than GenRL's complex sequence matching. FOUNDER then avoids the reward hacking problem, as the trends of the two return curves are now generally consistent, as shown in Table 1. The incorporation of TempD-based rewards enables FOUNDER to achieve superior performance on dynamic tasks such as Walk and Run, compared to FOUNDER w/o TempD and GenRL.

Moreover, Table 1 shows that incorporating TempD will not only benefit temporal dynamic tasks, but also match or surpass the performance of FOUNDER w/o TempD on 8 of the 9 static tasks (Stand and Kitchen tasks). This demonstrates that the positive influence of TempD-based rewards is consistent.

*Table 8.* Evaluation of the consistency between learned pseudo rewards and ground-truth rewards for different tasks in DMC.

| Methods | Corr↑ | Regret↓ | Precision↑ | Recall↑ | F1↑ |
|---|---|---|---|---|---|
| **cheetah run** | | | | | |
| GenRL | 0.49 | 0.19 | 0.50 | 0.77 | 0.60 |
| WM-CLIP | **0.80** | 0.42 | 0.57 | **1.00** | **0.73** |
| GenRL-TempD | 0.23 | 0.22 | 0.40 | 0.52 | 0.44 |
| FOUNDER w/o TempD | -0.05 | 1.11 | 0.00 | 0.00 | 0.00 |
| FOUNDER | 0.61 | **0.01** | **1.00** | 0.52 | 0.69 |
| **walker stand** | | | | | |
| GenRL | -0.11 | 0.12 | 0.77 | 0.53 | 0.63 |
| WM-CLIP | **0.78** | 0.02 | 0.81 | 1.00 | 0.89 |
| GenRL-TempD | -0.34 | 0.12 | 0.70 | 0.15 | 0.24 |
| FOUNDER w/o TempD | **0.78** | **0.00** | **1.00** | **0.88** | **0.94** |
| FOUNDER | -0.11 | 0.09 | **1.00** | 0.02 | 0.04 |
| **walker walk** | | | | | |
| GenRL | 0.11 | 0.22 | 0.52 | 0.29 | 0.37 |
| WM-CLIP | **0.65** | **0.00** | 0.97 | **0.67** | **0.79** |
| GenRL-TempD | 0.09 | 1.21 | 0.43 | 0.33 | 0.37 |
| FOUNDER w/o TempD | -0.20 | 1.56 | 0.00 | 0.00 | 0.00 |
| FOUNDER | 0.55 | 0.12 | **1.00** | 0.33 | 0.50 |
| **walker run** | | | | | |
| GenRL | **0.77** | 0.09 | 0.95 | 0.85 | **0.90** |
| WM-CLIP | 0.71 | 0.07 | 0.82 | **1.00** | **0.90** |
| GenRL-TempD | 0.12 | 0.96 | 0.67 | 0.49 | 0.56 |
| FOUNDER w/o TempD | -0.19 | 0.97 | 0.00 | 0.00 | 0.00 |
| FOUNDER | 0.58 | **0.04** | **1.00** | 0.33 | 0.50 |
| **stickman stand** | | | | | |
| GenRL | 0.23 | 0.24 | 0.08 | 0.17 | 0.11 |
| WM-CLIP | -0.36 | 0.86 | 0.07 | 0.14 | 0.09 |
| GenRL-TempD | 0.06 | 0.95 | 0.07 | 0.11 | 0.09 |
| FOUNDER w/o TempD | 0.27 | 0.79 | 0.10 | 0.17 | 0.12 |
| FOUNDER | **0.54** | **0.08** | **1.00** | **0.72** | **0.84** |
| **stickman walk** | | | | | |
| GenRL | -0.46 | 1.19 | 0.43 | **0.53** | 0.47 |
| WM-CLIP | 0.02 | 0.28 | 0.77 | 0.50 | **0.61** |
| GenRL-TempD | -0.14 | 1.24 | 0.58 | 0.42 | 0.49 |
| FOUNDER w/o TempD | -0.64 | 1.33 | 0.00 | 0.00 | 0.00 |
| FOUNDER | **0.82** | **0.10** | **1.00** | 0.41 | 0.59 |
| **stickman run** | | | | | |
| GenRL | -0.21 | 0.55 | 0.00 | 0.00 | 0.00 |
| WM-CLIP | 0.23 | 0.13 | 0.29 | 0.62 | 0.39 |
| GenRL-TempD | 0.36 | 0.56 | 0.39 | **0.98** | 0.56 |
| FOUNDER w/o TempD | -0.15 | 0.57 | 0.00 | 0.00 | 0.00 |
| FOUNDER | **0.82** | **0.02** | **1.00** | 0.95 | **0.97** |

