# OpenReview forum: "FOUNDER: Grounding Foundation Models in World Models for Open-Ended Embodied Decision Making"
_ICML.cc/2025/Conference — ICML 2025 poster_

### Official Review · Reviewer_bGqN · 2025-03-07

**Overall Recommendation:** 4

**Summary:**

FOUNDER proposes a method that leverages the generalization capability of world model dynamics alongside the prior knowledge embedded in foundation models to improve embodied decision making, and demonstrate its effectiveness through extensive experiments across diverse domains.

**Claims And Evidence:**

The claims made in the submission are supported by clear and convincing evidence; further details are discussed in the following section.

**Essential References Not Discussed:**

This paper appropriately cites and discusses the prior works necessary to understand and explain the proposed method.

**Experimental Designs Or Analyses:**

I checked the novelty of this submission and discussed the details in the section addressing its strengths and weaknesses.

**Methods And Evaluation Criteria:**

The proposed methods are appropriately evaluated using diverse criteria in environments such as Minecraft and DMC.

**Other Comments Or Suggestions:**

No

**Other Strengths And Weaknesses:**

- strength:
    - The authors conduct extensive experiments across multiple domains, demonstrating the approach’s robustness and scalability.
    - The selection of baselines appears well-reasoned, and the chosen metrics for comparison seem appropriate for evaluating the method’s improvements.
- weakness:
    - It seems that FOUNDER can be viewed as essentially LEXA [1] plus Foundation Model : LEXA already utilizes a similar world model framework along with temporal distance prediction. While FOUNDER’s approach and results appear compelling, it is less clear how novel the core idea is compared to existing solutions.
    - The key hyperparameters (e.g., the frame window size k, the KL weight, etc.) are not clearly summarized, and the paper would benefit from an ablation study detailing how these values affect performance.

**Questions For Authors:**

- In the behavior learning phase, it’s unclear in the paper how goal sampling is done during policy training. Do you randomly pick from the offline dataset, take one as the goal?
- If you’re calculating temporal distance with MSE as in Equation (7), wouldn't there be an issue if, for instance, the robot is standing still in the dataset or exhibits periodic behavior, resulting in incorrect learning?
- Since you’re using a foundation model, you really need to demonstrate generalization in the experiments. Does it still work if you query out-of-distribution goal images or prompts that weren’t used in training?

**Relation To Broader Scientific Literature:**

The paper provides a thorough analysis of the strengths and weaknesses of both existing world models and foundation models, proposing a method that leverages their complementary benefits to enhance embodied decision-making performance.

**Theoretical Claims:**

The mathematical formulations are well-structured and clearly presented, making them easy to understand.

---

> ### Author Rebuttal · Authors · 2025-04-01
>
> We sincerely thank the reviewer for their insightful comments and valuable feedback on our paper. We are delighted to receive recognition of our method's effectiveness, robustness, and scalability through extensive experiments, and for acknowledging that our claims are supported by convincing evidence with well-justified evaluation criteria (baselines/metrics). Please find below our detailed responses to the reviewer’s comments.
>
> **1. Novelty concerns:** We appreciate the reviewer's comparison between FOUNDER and LEXA. While both methods utilize world models and temporal distance prediction, their technical contributions differ fundamentally. LEXA focuses on discovering diverse goals and learning to achieve them during online exploration, enabling adaptation to image goals for downstream tasks. In contrast, the core idea of FOUNDER is to ground foundation model (FM) representations (from multimodal tasks) into the world model (WM)'s latent space through explicit mapping functions, operating entirely offline. This enables physics-consistent translation of VLM embeddings into actionable WM states. It should be noted that world model and temporal distance predictor in FOUNDER only serve as tools to implement this paradigm, whereas the core contribution resides in bridging the gap between FM's general knowledge with WM's physical dynamics, establishing a new methodology for open-ended task solving and setting FOUNDER apart from prior methods using VLMs in RL.
>
> **2. Hyperparameter ablation**: We appreciate the reviewer's feedback regarding the hyperparameter settings. For frame window size (k=8), this choice strictly aligns with two critical considerations: (1) InternVideo2's pretrained temporal encoder requires fixed 8-frame inputs, as validated in its implementation. (2) Retained GenRL's 8-frame configuration for fair comparison. For the kl weight β, we use β=1.0 by default. To address your concerns, we conducted an ablation study on the Cheetah and Kitchen domains to validate their sensitivity, and the results are on [our website](https://sites.google.com/view/founder-rl). We find that FOUNDER is not sensitive to the kl weight parameter, while 1.0 is the best choice.
>
> **3. Goal sampling in behavior learning**: In fact, the goal state zg of a given video/text task during behavior learning is sampled from the goal distribution inferred from the video/text VLM embedding, e.g., using the learned mapping function: zg∼Qψ(⋅∣eg). This is discussed in Equation (6) and Appendix C.1.
>
> **4. Question about temporal distance learning**: We thank the reviewer for raising this valuable point. Learning a temporal distance predictor from quasi-static data may pose risks. However, the world model provides proper regularization to the input states, and such datasets are rare in real applications. To address your concerns, we provide experiments on learning a temporal distance predictor on the Stand datasets, where trajectories are generated by the expert policy in Walker Run, and validate the predictor's accuracy in predicting the same WM states as near-zero temporal distance. The results on [our website](https://sites.google.com/view/founder-rl) show robustness of temporal prediction in such settings.
>
> **5. Out-of-distribution tasks**: In section 5.2, we evaluate the performance of FOUNDER on out-of-distribution video tasks, where the visual appearance of task video prompts differs significantly from the agent's environment observations and training data, including cross-embodiment and cross-viewpoint settings on DMC and Kitchen, and FOUNDER demonstrates superior performance than other baselines. Also, in Sections 5.1 and 5.4, we test FOUNDER on out-of-distribution language tasks like Cheetah Flipping and Hunt Sheep, where the training dataset does not contain buffers of these tasks, and the performance of FOUNDER exhibits its generalization abilities. To further address your concerns, we present experimental results on real-world video tasks on [our website](https://sites.google.com/view/founder-rl) using videos provided in GenRL's code repository, and FOUNDER also demonstrates solid performance compared to GenRL.

---

### Official Review · Reviewer_GtTJ · 2025-03-14

**Overall Recommendation:** 3

**Summary:**

This paper introduces FOUNDER, a framework for grounding foundation models (FMs) in world models (WMs) to enable generalizable representation learning, multi-modal prompting, and dense reward prediction. FOUNDER shares conceptual similarities with GenRL but incorporates temporal information to enhance reward prediction and goal-conditioned reinforcement learning (RL) for more flexible behavior cloning. Experimental results on DMC and Kitchen benchmarks demonstrate that FOUNDER outperforms GenRL baselines.

## update after rebuttal

The author's response addressed my concerns. I would like to remain my original ratings.

**Claims And Evidence:**

Yes.

**Essential References Not Discussed:**

None.

**Experimental Designs Or Analyses:**

The experimental design appears sound.

**Methods And Evaluation Criteria:**

Yes.

**Other Comments Or Suggestions:**

None.

**Other Strengths And Weaknesses:**

Strength
- The paper is clearly written and well-organized.
- The experimental results effectively demonstrate the advantages of the proposed FOUNDER framework.

Weakness
- The proposed temporal distance prediction does not consistently improve performance.
  - For example, in Kitchen Burner (Table 1), the mean score drops significantly from 1.0 to 0.6.
  - Similar performance degradation is observed in the Minecraft environment, where Figure 5 shows that FOUNDER w/o TempD consistently outperforms FOUNDER.
  - It remains unclear whether temporal distance prediction is a broadly applicable method or merely a task-specific trick. The paper does not provide sufficient analysis regarding: 1) Which types of tasks benefit from temporal distance prediction? 2) Under what conditions does it fail? 3) What are the underlying reasons for performance degradation in certain tasks?

**Questions For Authors:**

Please refer to the Weaknesses section.

**Relation To Broader Scientific Literature:**

FOUNDER shares similarity with GenRL but introduces temporal distance prediction and a goal-conditioned RL framework, resulting in more robust performance. It also relates to WM-CLIP, which learns a mapping from WM states to FM representations. However, FOUNDER instead grounds FM representations in the WM state space, leading to improved performance.

**Theoretical Claims:**

NA.

---

> ### Author Rebuttal · Authors · 2025-04-01
>
> We sincerely thank the reviewer for their valuable feedback on our paper. We are delighted to receive recognition of our method's sound experimental design and solid experimental results. Please find below our detailed responses to the reviewer’s comments.
>
> **The influence of temporal distance**: We appreciate the reviewer for raising these important questions regarding temporal distance (TempD). Generally, the role of TempD-based rewards is to enhance temporal awareness and provide task-completion information to the FOUNDER agent.
>
> From Table 1, using cosine similarity between VLM representations of task and observations as rewards, FOUNDER w/o TempD already performs well on static tasks (Stand tasks and Kitchen tasks). However, FOUNDER w/o TempD struggles in dynamic tasks (Walk, Run). We observe that in dynamic tasks, the agent can perform well at the very beginning of the behavior learning stage, but its performance deteriorates as training progresses, eventually reaching very low performance by the end. We also discovered that the real return and pseudo return curves exhibit completely opposite trends during behavior learning. The real performance worsens as the agent maximizes the pseudo reward, causing a reward hacking problem. We then visualize the resulting trajectories of the policy at the early stage and the final policy. Despite performing well at the beginning, we find that at the end, the agent’s behavior becomes more static. The agent may stay at its initial position, appearing as if it is running or walking, but in reality, it is only 'running' or 'walking' at the original place without progressing forward, or moving forward at an extremely slow pace. This explains the poor final performance of FOUNDER w/o TempD in Table 1. These training curves, trajectory visualization, and detailed analysis can be found on [our website](https://sites.google.com/view/founder-rl).
>
> We conclude that reward functions based on cosine similarity or other direct distance metrics may lead the policy to mimic the visual appearance of the goal, while overlooking the underlying task semantics and multi-step movement, particularly in dynamic tasks like running or walking. Since the FOUNDER-based method maps the target sequence to a single goal state in the world model, we hypothesize that using direct distance functions may result in a lack of temporal awareness.
>
> At this time, the role of TempD-based rewards is to provide temporal information and crucial task-completion information to the agent, with a much lower training cost than GenRL's complex sequence matching. In Table 1, FOUNDER then avoids the reward hacking problem and achieves superior performance on dynamic tasks like Walk and Run, compared to GenRL and FOUNDER w/o TempD.
>
> Moreover, Table 1 shows that incorporating TempD will not only benefit temporal dynamic tasks but also match or surpass the performance of FOUNDER w/o TempD on static tasks. Kitchen Burner is the only 1 of the 9 static tasks in Table 1 where FOUNDER is outperformed by FOUNDER w/o TempD, which does not constitute statistical significance.
>
> Furthermore, for Minecraft tasks, FOUNDER w/o TempD performs surprisingly good, and incorporating TempD could weaken the performance on several tasks. We hypothesize that the data quality and the stochasticity nature of the Minecraft environment may have a negative impact on the learning process of the TempD predictor. When temporal predictions become noisy, falling back to cosine similarity serves as a safer baseline, and using visual similarities as pseudo-rewards in short-horizon Minecraft tasks is beneficial and will not cause problems on Walk or Run tasks in DMC. However, when TempD learns accurate temporal dynamics, its temporal credit assignment mechanism provides provable advantages in reward-free multi-task RL, as is proved in prior works [1-4]. We will incorporate approaches discussed in our rebuttal with Reviewer 2yAX, as well as techniques in [1], for better TempD learning under stochastic and complex environments.
>
> [1] Myers V, Zheng C, Dragan A, et al. Learning Temporal Distances: Contrastive Successor Features Can Provide a Metric Structure for Decision-Making[C]//International Conference on Machine Learning. PMLR, 2024: 37076-37096.
>
> [2] Park S, Kreiman T, Levine S. Foundation Policies with Hilbert Representations[C]//International Conference on Machine Learning. PMLR, 2024: 39737-39761.
>
> [3] Park S, Rybkin O, Levine S. Metra: Scalable unsupervised rl with metric-aware abstraction[J]. arXiv preprint arXiv:2310.08887, 2023.
>
> [4] Mendonca R, Rybkin O, Daniilidis K, et al. Discovering and achieving goals via world models[J]. Advances in Neural Information Processing Systems, 2021, 34: 24379-24391.

---

> > ### Comment · Reviewer_GtTJ · 2025-04-02
> >
> > The authors' response addressed most of my concerns. I will maintain my original score.

---

> > > ### Author Response · Authors · 2025-04-04
> > >
> > > We are glad to hear that our efforts to address your concerns have been well-received. Thank you for your time and consideration.

---

### Official Review · Reviewer_2yAX · 2025-03-14

**Overall Recommendation:** 3

**Summary:**

This work proposes FOUNDER, a method that leverages Visual-Language Models (VLMs) to get representations from visual observations and train RL agents on World Models imagination, starting from an offline dataset of trajectories. It proposes a method of aligning the embeddings from the VLM with the latent states learned by the World Model via a learned mapping function. This mapping function is optimized to reconstruct the VLM embeddings while also covering the WM latent distribution (via KL divergence minimization). For behavior learning, FOUNDER first learns a temporal distance predictor model, which is then used as reward model for the imagined rollouts. The agent then is learned via actor-critic RL in these imagined trajectories. The work presents experiments in DMC/Kitchen/Minecraft environments, claiming improvements over closer baselines (GenRL and WM-CLIP [1]).

**Claims And Evidence:**

I found two core claims from the paper:

1) The proposed method improves the alignment between the VLMs representations and the WM representations, which allows it to capture the semantic of the tasks better. The evidence would be the downstream results in the DMC/Kitchen/Minecraft environments. While I agree this is a proper form of evidence, I have some concerns on the experimental design (to be described later)

2) The temporal distance predictor provides a more consistent and informative reward signal, while prior methods are prone to reward hacking. The presented evidence is on qualitative analysis of failure cases and correlation analysis between the proxy reward and ground truth reward.

There is one additional claim, which is not central as the previous ones, but still debatable: The proposed components of the method are task-agnostic and therefore “universally applicable to any downstream task and effectively facilitating open-ended task solving in embodied environments”. While it is clear that the modeling assumptions are task-agnostic, I think it is too strong to claim that they are universally applicable to any downstream task as there is no evidence these components would actually generalize/learn any task and also it is unclear how much this actually relies on the properties of the offline dataset.

**Essential References Not Discussed:**

I couldn’t find any essential references not discussed.

**Experimental Designs Or Analyses:**

I have some major concerns in terms of the experimental design.

- The presented numbers (Table 1) does not reproduce what was claimed in GenRL paper [1]. In fact, there are substantial changes in some environments. It is unclear why this happens. Is the offline dataset used in this work different? If so, why? Given these discrepancies, it is hard to tell if GenRL was fully reproduced or if these methods are too sensitive for the employed offline dataset.

- The paper does not bring any offline RL method as baseline, arguing that it involves a “cumbersome process for retraining from scratch for every task”. While I understand this, I believe they are necessary here to justify the need for a more complex, model-based approach. Furthermore, this is the standard in prior work as well - for instance, GenRL does bring these methods, which clarifies this point.

- The minecraft experiments look very inconclusive. The standard errors are too spread for all methods in a way that is hard to claim that FOUNDER-based methods consistently outperform GenRL with statistical significance.  This is concerning since it is supposed to be the most challenging benchmark evaluated in the work, and the gains are very unclear.

**Methods And Evaluation Criteria:**

I believe that the proposed method, considered baselines, and evaluation criteria make sense. Nonetheless, I am not sure if the described problem setting and parts of the narrative really align. More concretely, the problem setting describes as a goal “solving open-ended tasks in the context of offline reinforcement learning from visual observations”, and it is unclear the open-ended terminology here, which is often related to interestingness (information gain) and learnability [2]. From what was presented in the paper, it is not clear how a method in a pure offline setting, with a clear definition of goals (as explicitly model as goal conditioned policy) would be learning open-ended behavior.


Apart from that, there are some design choices which are questionable/unjustified (See weaknesses below)

**Other Comments Or Suggestions:**

N/A

**Other Strengths And Weaknesses:**

I believe the paper should discuss more some potential limitations:

1) Limitations related to the temporal distance predictor:

    - The temporal distance predictor relies on a predefined sequence length T, but on test-time this is not available (unless the problem setting is limited to fixed-horizon episodic tasks, but this is not specified).

    - Also, why isn’t the temporal-distance predictor reward model also not prone to reward hacking? In a harder environment I would expect it to misgeneralize and present the same effect.

    - There is also a potential limitation on mining negative examples for the temporal distance predictor training, which the paper does not explore and only performs randomly.

2) The “+1 reward” is not justified/discussed in proper depth. As a simple reward shaping, I would not expect to have such a big impact in the final performance as presented in Table 5. It would be very important to discuss the effect of this heuristic.

3) One last thing, while not specific to FOUNDER, but more broadly for these reward-free, offline settings: there is a strong reliance on the properties of the employed offline dataset. I believe this should be discussed in a limitations sections, cont

**Questions For Authors:**

Please see my questions in the previous points.

**References**

[1] Mazzaglia et. al. GenRL: Multimodal-foundation world models for generalization in embodied agents. NeurIPS, 2024.

[2] Hughes et. al. Position: Open-Endedness is Essential for Artificial Superhuman Intelligence. ICML, 2024.


**Please refer to my rebuttal comment for the updated score**

**Relation To Broader Scientific Literature:**

In my perspective, the main contribution of this work is to identify and highlight that the reward modeling aspect of GenRL leads to reward misspecification and hacking. The proposed method for alignment embeddings also looks relevant, although it is also unclear if this really works better than GenRL method (from Table 1, GenRL works better than Founder w/o TempD).

**Theoretical Claims:**

No theoretical claims.

---

> ### Author Rebuttal · Authors · 2025-04-01
>
> We sincerely thank the reviewer for their insightful comments and valuable feedback. Below are our responses to each of their concerns:
>
> **Universally applicable claim**: Our claim of task-agnosticism refers to FOUNDER’s architecture—world model, mapping function, and reward generator—enabling deployment across downstream tasks specified via text or video inputs without architectural changes. This highlights its broad applicability, not guaranteed universal performance. We appreciate your feedback and will clarify this in the final version.
>
> **Open-endedness**: In our context, “open-endedness” refers to the agent’s ability to ground and solve user-defined tasks from free-form prompts via multimodal interfaces, without predefined reward functions. This differs from definitions emphasizing self-evolving exploration or unbounded learnability. FOUNDER achieves this capability by jointly learning the WM, mapping function, reward generator, and downstream policy from offline data. This definition aligns with similar works [1-3]. We will clarify this in the final version.
>
> **Performance of GenRL**: Our reproduction of GenRL follows its official code and datasets. Our reproduced results (GenRL_ours) align with GenRL’s arXiv v1 metrics on in-distribution DMC tasks but underperform on Kitchen tasks, likely due to domain properties. Despite GenRL v2’s improved performance, its implementation was not updated. FOUNDER outperforms GenRL in both cases.
>
> |  | GenRL_v1 (reported) | GenRL_v2 (reported) | GenRL_ours (reproduced) | FOUNDER |
> | --- | --- | --- | --- | --- |
> | **DMC** | 0.74 $\pm$ 0.02 | 0.82 $\pm$ 0.01 | 0.75 $\pm$ 0.08 | 0.87 $\pm$ 0.03 |
> | **Kitchen** | 0.69 $\pm$ 0.15 | 0.76 $\pm$ 0.05 | 0.50 $\pm$ 0.32 | 0.89 $\pm$ 0.17 |
>
> **Effectiveness of FOUNDER's components:** Our key innovation is grounding VLM representations into WM states without GenRL’s costly sequence matching, enhancing task understanding beyond reward correction. As shown in Table 1, FOUNDER w/o TempD outperforms GenRL on static tasks (e.g., stand, Kitchen), while it requires TempD for dynamic tasks (e.g., Walk, Run) to improve temporal awareness. TempD efficiently provides temporal structure at a much lower computational cost than GenRL’s sequence matching. TempD’s addition to GenRL does not improve its performance, confirming FOUNDER’s architecture drives success.
>
> **The “+1 reward” heuristic**:  The +1 operation smooths rewards, improving learning. Experiments show agents can achieve similar results without this shaping given more training steps. Training curves comparing shaped and unshaped rewards are available on [our website](https://sites.google.com/view/founder-rl).
>
> **Minecraft Experiment**: High variance in Minecraft experiments is mainly due to the environment’s stochasticity. To better reflect statistical significance, we now report 95% confidence intervals and provide clearer learning curves on our website. FOUNDER matches GenRL on 2/5 tasks and outperforms it on the remaining three.
>
> **Comparison with model-free baselines**: We directly compare **HILP**, a multi-task model-free approach, and **TD3**, the strongest single-task model-free baseline per GenRL. As detailed in our response to reviewer e54P, these comparisons justify the need for a model-based approach, with FOUNDER achieving a strong balance between performance and efficiency.
>
> **Limitations related to the temporal distance predictor**:
>
> 1. The predictor learns with a predefined sequence length during training, but during testing, it outputs the predicted temporal distance independent of sequence length.
> 2. Temporal distance excels in cross-domain tasks and metric evaluation, where GenRL and cosine similarity methods may fail due to reward hacking. While extreme OOD scenarios may still pose challenges, FOUNDER’s success shows its robustness in capturing deep task semantics rather than relying on brittle visual correlations.
> 3. We appreciate the reviewer’s insight on mining negative samples. We follow LEXA [4], but acknowledge the risk of mislabeling. This can be mitigated by filtering out high-similarity negative pairs or actively selecting challenging cross-sequence pairs via feature-space searching. We will include an ablation on this in the final version.
>
> **Limitations from offline data quality**: We agree with the reviewer that dataset properties limit reward-free offline RL methods and have discussed this in Section 6. We welcome further discussion on dataset dependency concerns, as the reviewer did not fully elaborate on the complete questions.
>
> [1] Fan L, et al. Minedojo: Building open-ended embodied agents with internet-scale knowledge. NeurIPS 2022.
>
> [2] Team O E L, et al. Open-ended learning leads to generally capable agents. arXiv preprint.
>
> [3] Qin Y, et al. Mp5: A multi-modal open-ended embodied system in minecraft via active perception. CVPR 2024.
>
> [4] Mendonca R, et al. Discovering and achieving goals via world models. NeurIPS 2021.

---

> > ### Comment · Reviewer_2yAX · 2025-04-04
> >
> > Thank you for your rebuttal. After careful examination, I can say that most of my concerns were addressed.  I still believe all the limitations I raised are valid and I strongly recommend the work to discuss them.  Furthermore, while the "+1 reward" is not necessary to attain the best performance (and brought in the new results), it still does make a big difference, and the justification given is still vague.
> >
> > I also strongly recommend authors to incorporate the feedback on the claims about universally applicability and open-endedness, otherwise the work may sound misleading for some audience.
> >
> > Nonetheless, I don't think these points are grounds for rejection, so I am raising my score to 3.

---

> > > ### Author Response · Authors · 2025-04-08
> > >
> > > We sincerely thank the reviewer's feedback and the updated score. We are glad to hear that our efforts to address most of your concerns have been well-received. We appreciate your valuable suggestions, and we will ensure a thorough discussion and clarification of the points you raised, such as task-agnostic applicability and open-ended multi-modal task-solving capability, in the final version of our work.
> > >
> > > Moreover, regarding the "+1 reward", we have released training curves comparing performance with and without the "+1 reward" on our website. These results confirm that agents can eventually achieve similar performance without the "+1 shaping" over more training steps, demonstrating that the "+1" operation is an optional engineering choice rather than a fundamental component of our method.
> > >
> > > We acknowledge that the "+1" shaping does improve learning efficiency and stability. Since most of the originally predicted temporal distances are clustered near -1, the "+1" operation shifts the rewards from near -1 to near 0, and it is common for prior works [3-5] to use zero-centering rewards (e.g. clipping or rescaling rewards to be zero-centered) to enhance learning. Additionally, the "+1" operation is a theoretically grounded instance of potential-based reward shaping [6] that preserves policy optimality, where the potential function induces a constant reward shift.
> > >
> > > Furthermore, directly normalizing the original temporal distance (without "+1") and using the resulting rewards yields similar performance and learning efficiency to the "+1" reward, as shown in the new learning curves results on [our website](https://sites.google.com/view/founder-rl). This indicates that the positive impact of the "+1" operation on the performance is similar to that of reward normalization.
> > >
> > > In conclusion, the "+1" operation and normalization are merely reward shaping techniques. While these shaping methods may affect learning performance [1-2], they are not central to the core of our approach. Nevertheless, we sincerely thank the reviewer for raising and discussing this issue, as it provides us with valuable insights.
> > >
> > > We deeply appreciate the reviewer’s time, consideration, and constructive feedback.
> > >
> > > [1] Henderson P, Islam R, Bachman P, et al. Deep reinforcement learning that matters[C]//Proceedings of the AAAI conference on artificial intelligence. 2018, 32(1).\
> > > [2] Van Hasselt H P, Guez A, Hessel M, et al. Learning values across many orders of magnitude[J]. Advances in neural information processing systems, 2016, 29.\
> > > [3] Hessel M, Modayil J, Van Hasselt H, et al. Rainbow: Combining improvements in deep reinforcement learning[C]//Proceedings of the AAAI conference on artificial intelligence. 2018, 32(1).\
> > > [4] Van Hasselt H, Guez A, Silver D. Deep reinforcement learning with double q-learning[C]//Proceedings of the AAAI conference on artificial intelligence. 2016, 30(1).\
> > > [5] Andrychowicz M, Wolski F, Ray A, et al. Hindsight experience replay[J]. Advances in neural information processing systems, 2017, 30.\
> > > [6] Ng A Y, Harada D, Russell S. Policy invariance under reward transformations: Theory and application to reward shaping[C]//Icml. 1999, 99: 278-287.

---

### Official Review · Reviewer_e54P · 2025-03-17

**Overall Recommendation:** 3

**Summary:**

The paper proposed FOUNDER, a novel framework that integrates Foundation Models (FMs) with World Models (WMs) to enable reward-free, open-ended decision-making in embodied environments. The central idea is to ground FM representations into the WM state space, allowing GCRL through imagination. Instead of relying on manually crafted reward functions, FOUNDER estimates the temporal distance to goal states as an intrinsic reward signal, leading to superior task generalization. The proposed method is evaluated on multi-task offline visual control benchmarks, including the DeepMind Control Suite, Franka Kitchen, and Minecraft, demonstrating strong performance in learning task semantics from text or video prompts, especially in challenging cross-domain settings. Empirical results show that FOUNDER significantly outperforms prior methods like GenRL by leveraging deeper semantic understanding rather than relying on step-by-step visual alignment.

**Claims And Evidence:**

The paper claims that FOUNDER enhances task generalization by bridging FM knowledge with WM-based decision-making, which is supported by empirical results showing higher success rates across diverse domains. Another claim is that the temporal distance-based reward function provides a more reliable training signal than traditional visual similarity metrics, validated through improved reward consistency analysis. The authors also claim that FOUNDER performs well in cross-domain tasks, where large domain gaps exist between task prompts and the embodied environment. This is demonstrated by its superior performance in generalizing across different camera viewpoints and agent embodiments in tasks like cheetah and Minecraft.

**Essential References Not Discussed:**

No

**Experimental Designs Or Analyses:**

The experiments are extensive, with evaluations on both text-based and video-based task prompts across multiple simulated environments.

**Methods And Evaluation Criteria:**

The evaluation protocol covered both locomotion and manipulation tasks with multiple baselines, including GenRL, WM-CLIP, and ablations of FOUNDER without temporal distance prediction. Except there are no real-world applications, the experiments are extensive and detailed.

The author talked about efficiency or inefficiency at a few places, so maybe a quantitative result and analysis for efficiency would be more convincing.

**Other Comments Or Suggestions:**

No

**Other Strengths And Weaknesses:**

The paper is well-written and straightforward to read, and the experiments are extensive and solid. One major concern is I am not sure how the proposed method compared with other methods, especially the model-free method, in both the performance, efficiency and even real-world applications. I noticed the author stated that "Model-free offline RL methods like IQL (Kostrikov et al., 2021b) and TD3+BC (Fujimoto & Gu, 2021) involve the cumbersome process of retraining from scratch for every task, and zero-shot FM-generated rewards using task prompts and observations has been shown to perform poor in (Mazzaglia et al., 2024). Therefore, we compare FOUNDER only with model-based methods." I may hold a more conservative opinion on this: for the "cumbersome process of retraining from scratch for every task" I believe there are lots of multi-task on-policy work that may also be a good and fair comparison, and seemed the published date of 2021 is far more behind 2024. Nevertheless, requiring intensive extra experiments for model-free methods may be hard in the rebuttal phase, I may expect more analysis on the efficiency part.

**Questions For Authors:**

All experiments primarily focus on tabletop and 2D short-horizon tasks. Given this, do the authors anticipate that FOUNDER’s integration of  WM could be even more beneficial for long-horizon tasks that require memory and temporal abstraction? Specifically, could the WM’s ability to model latent state transitions and predict future states provide an advantage in tasks where agents must recall and utilize past information over extended time horizons? If so, what modifications or enhancements might be necessary for FOUNDER to effectively scale to such settings?

**Relation To Broader Scientific Literature:**

The paper is well-positioned within the literature on foundation models for reinforcement learning, goal-conditioned RL, and world models. It builds upon prior work such as GenRL, DreamerV3, and Choreographer, differentiating itself by focusing on task grounding through state-space alignment.

**Theoretical Claims:**

There are no theoretical claims.

---

> ### Author Rebuttal · Authors · 2025-04-01
>
> We sincerely thank the reviewer for their insightful comments and valuable feedback on our paper. We are delighted to receive recognition of our method's solid performance and strong task generalization demonstrated by extensive and detailed experiments. Please find below our detailed responses to the reviewer’s comments.
>
> **Other model-free baselines**:
> Prior work (e.g., GenRL) has proven that traditional model-free offline RL methods (IQL, TD3+BC, TD3) perform poorly in this setting compared to model-based approaches. These methods rely on zero-shot pseudo-rewards generated by Foundation Models (FM) through cosine similarity between FM representations of task prompts and observations. However, FM rewards are not explicitly grounded in the embodied domain, limiting their effectiveness. Additionally, traditional model-free methods require training from scratch for each task, making them inefficient and impractical for multi-task solving.
>
> However, to ensure a comprehensive evaluation and address your concerns, we now include HILP [1], a multi-task model-free method that achieves strong zero-shot RL performance in goal-conditioned RL, and TD3, the best-performing single-task model-free baseline according to GenRL’s experiment. We assess their performance across eight tasks in the Cheetah and Kitchen domains, alongside GenRL for a clearer comparison. The results are on [our website](https://sites.google.com/view/founder-rl). We find that FOUNDER consistently outperforms both single-task and multi-task model-free baselines, and our prior claim is reinforced. We sincerely appreciate the reviewer’s suggestion regarding model-free baseline comparisons.
>
> **Efficiency and computational costs**:
> We appreciate the reviewer's emphasis on computational efficiency and provide a comparison of the four typical approaches mentioned. All experiments were conducted on an RTX 3090 GPU. For pretraining, training the MFWM in GenRL for 500k gradient steps requires approximately five days, whereas FOUNDER reduces this overhead to about three days by avoiding the seq2seq-style generative sequence modeling used in GenRL. Pretraining the Hilbert representation and foundation policy in HILP for 500k gradient steps takes around 30 hours.
>
> For downstream tasks, training the actor-critic on a given task prompt for 50k gradient steps takes under five hours for both GenRL and FOUNDER. In the case of model-free methods, HILP enables zero-shot adaptation to downstream tasks, whereas single-task model-free methods like TD3 require approximately seven hours to train from scratch for 500k gradient steps. Overall, we find that FOUNDER strikes a strong balance between performance and learning efficiency.
> |   | TD3 | HILP | GenRL| FOUNDER |
> | :--- | :--- | :--- |:--- | :--- |
> | **Performance** | Low | Low | Medium | High |
> | **Multi-Task Adaptation**| From scratch (~7h) | Zero-shot | Finetune (~5h) | Finetune (~5h) |
> | **Pretraining Overhead**| N/A | ~30h | ~120h | ~72h |
> | **Overall Computational Cost**| Low | Medium-Low | High | Medium |
>
>
> **Scalability of FOUNDER**
> We have included a brief discussion on the scalability of FOUNDER to long-horizon tasks in Section 6. While our current experiments focus on short-horizon tasks, FOUNDER’s architecture is inherently designed to support long-horizon reasoning through two key mechanisms: (1) the world model’s temporal abstraction via latent state transitions and (2) the foundation model’s semantic grounding, which enables hierarchical task decomposition. To fully unlock this potential, there may exist three enhancements: (i) integrating more powerful sequence modeling architectures (e.g., [2]) or continual learning methods (e.g., [3]) a for persistent context tracking, (ii) implementing FM-guided curriculum learning [3] to phase in complex tasks, and (iii) developing a meta-controller for dynamic subgoal generation. We thank the reviewer for raising this concern, and we will explore it in our future work.
>
> [1] Park S, Kreiman T, Levine S. Foundation Policies with Hilbert Representations[C]//International Conference on Machine Learning. PMLR, 2024: 39737-39761.\
> [2] Samsami M R, Zholus A, Rajendran J, et al. Mastering memory tasks with world models[J]. arXiv preprint arXiv:2403.04253, 2024.\
> [3] Feng T, Wang X, Zhou Z, et al. EvoAgent: Agent Autonomous Evolution with Continual World Model for Long-Horizon Tasks[J]. arXiv preprint arXiv:2502.05907, 2025.

---

> > ### Comment · Reviewer_e54P · 2025-04-08
> >
> > The authors' response addressed most of my concerns. I will maintain my original score.

---

> > > ### Author Response · Authors · 2025-04-08
> > >
> > > We are glad to hear that our efforts to address your concerns have been well-received. Thank you for your time and consideration.

---

### Decision · Program_Chairs · 2025-05-01

**Decision:**

Accept (poster)

**Comment:**

The authors introduce FOUNDER, a novel framework that integrates Foundation Models with World Models for reward-free decision-making in embodied environments by grounding visual-language model representations into the World Model state space through a learned mapping function. This approach enables goal-conditioned reinforcement learning with stronger generalization capabilities compared to traditional manually crafted reward functions. Experiments across DeepMind Control Suite, Franka Kitchen, and Minecraft demonstrate FOUNDER's superior performance over baselines like GenRL and WM-CLIP.

After the rebuttal process, the paper received positive scores [3,3,3,4], with all reviewers supporting acceptance. While using foundation models to provide self-supervised reward signals for reinforcement learning is not a new concept, FOUNDER's impressive experimental results clearly demonstrate the framework's significance and broad applicability to open-ended embodied AI.  Given these strengths and the consensus among reviewers, I recommend accepting this paper for publication.